# Volcanic crisis reveals coupled magma system at Santorini and Kolumbo

Marius P. Isken[1,14 ✉], Jens Karstens[2,14 ✉], Paraskevi Nomikou[3], Michelle Maree Parks[4], Vincent Drouin[4], Eleonora Rivalta[1,5], Gareth J. Crutchley[2], Mahmud Haghshenas Haghighi[6], Emilie E. E. Hooft[7], Simone Cesca[1], Thomas R. Walter[1,8], Sebastian Hainzl[1,8], Joachim Saul[1], Dimitris Anastasiou[9], Kostas Raptakis[9], Nikolai M. Shapiro[10], Jannes Münchmeyer[10], Quentin Higueret[10], Jean Soubestre[10], Florent Brenguier[10], Rebeckah S. Hufstetler[7], Kaisa R. Autumn[7], Maria Tsakiri[9], Dietrich Lange[2], Heidrun Kopp[2,11], Morelia Urlaub[2,11], María Blanch Jover[2], Jonas Preine[12], Christian Hübscher[13], Mahdi Motagh[1,10], Daniel Müller[1], Torsten Dahm[1,8] & Christian Berndt[2,11]

Volcanic crises, driven by renewed magma inflow and migration, result in surface deformation and seismicity that can provide unique insights into the structure of volcanic systems and magmatic processes. Although the highly explosive volcanoes of Santorini and Kolumbo[1,2] in the Greek Aegean Sea are just 7 km apart, their potentially coupled deep magmatic feeding systems are only poorly understood[3,4]. The 2025 volcano–tectonic crisis of Santorini simultaneously affected both volcanic centres, providing insights into a complex, multistorage feeder system. Here we integrate onshore and marine seismological data with geodetic measurements to reconstruct magma migration before and during the crisis. Gradual inflation in the Santorini caldera, beginning in mid-2024, preceded the January 2025 intrusion of a magma-filled dike sourced from a mid-crustal reservoir beneath Kolumbo, indicating a link between the two volcanoes. Joint inversion of ground and satellite-based deformation data indicates that approximately 0.31 km$^3$ of magma intruded as an approximately 13-km-long dike, reactivating principal regional faults and arresting 3–5 km below the seafloor. The 2024–2025 resurgence of magmatic activity beneath both volcanic centres and their apparent coupling provides insights into the dynamic interplay of magma storage, transport and reservoir failure beneath neighbouring volcanoes.

Volcanic eruptions are often preceded by magmatic dike intrusions, which manifest themselves in ground deformation, increased seismicity and changes in gas emissions as observed before the 2000 Miyake-jima (Japan), the 2018 Kilauea (Hawaii, USA), as well as the 2014–15 Bárðarbunga–Holuhraun and the 2021–24 Reykjanes Peninsula (Iceland) eruptions[5–10]. Monitoring these precursors is critical for detecting volcanic unrest, assessing hazards and providing early warnings of eruption-related hazards. Volcanic activity remains challenging to assess and forecast because of complex and cascading interactions. Therefore, innovative approaches and advances in integrating high-resolution earthquake locations with geodetic surface displacement data are becoming essential for reconstructing the dynamic processes in volcanic feeding systems.

Following elevated seismic activity centred on the Santorini caldera since September 2024 (Extended Data Fig. 1), an unexpectedly intense earthquake swarm began at about 19:00 UTC on 27 January 2025, located 10 km NE of Santorini and 2 km SE of Kolumbo (Figs. 1 and 2). The 2025 Santorini earthquake swarm continued with fluctuating intensity for more than 30 days. It exhibited a progressive energy release with a pattern of hypocentre migration focussing between the islands of Santorini, Anhydros and Amorgos (Fig. 1a). The seismic crisis led the Greek authorities to declare a state of emergency on Santorini and neighbouring islands between 6 February and 3 March 2025.

Santorini is part of the Christiana–Santorini–Kolumbo volcanic field. It is considered to be in a period of rejuvenation and magma replenishment following the Minoan eruption of 1600 BCE, one of the largest volcanic events of the Holocene[1,11]. Santorini and the neighbouring Kolumbo volcano have produced highly explosive, large-scale eruptions in historical times, including the Kameni eruption of 726 CE and the Kolumbo eruption of 1650 CE[1,2]. The volcanoes are located 7 km apart, thus within range of possible coupling[12]. Seismic velocity anomalies, repeated earthquake swarms beneath Santorini and Kolumbo

[1]GFZ Helmholtz Centre for Geosciences, Potsdam, Germany. [2]GEOMAR Helmholtz Centre for Ocean Research Kiel, Kiel, Germany. [3]Department of Geology and Geoenvironment, National and Kapodistrian University of Athens, Athens, Greece. [4]Icelandic Meteorological Office, Reykjavík, Iceland. [5]Section of Geophysics, Department of Physics and Astronomy, Alma Mater Studiorum University of Bologna, Bologna, Italy. [6]Institute of Photogrammetry and GeoInformation, Leibniz University Hannover, Hannover, Germany. [7]Department of Earth Science, University of Oregon, Eugene, OR, USA. [8]Institute of Geosciences, University of Potsdam, Potsdam, Germany. [9]School of Rural, Surveying and Geoinformatics Engineering, National Technical University of Athens, Zographos, Greece. [10]Université Grenoble Alpes, Université Savoie Mont Blanc, CNRS, IRD, Université Gustave Eiffel, ISTerre, Grenoble, France. [11]Institute of Geosciences, Kiel University, Kiel, Germany. [12]Department of Geology and Geophysics, Woods Hole Oceanographic Institution, Woods Hole, MA, USA. [13]Institute of Geophysics, University of Hamburg, Hamburg, Germany. [14]These authors contributed equally: Marius P. Isken, Jens Karstens. ✉e-mail: marius.isken@gfz.de; jkarstens@geomar.de

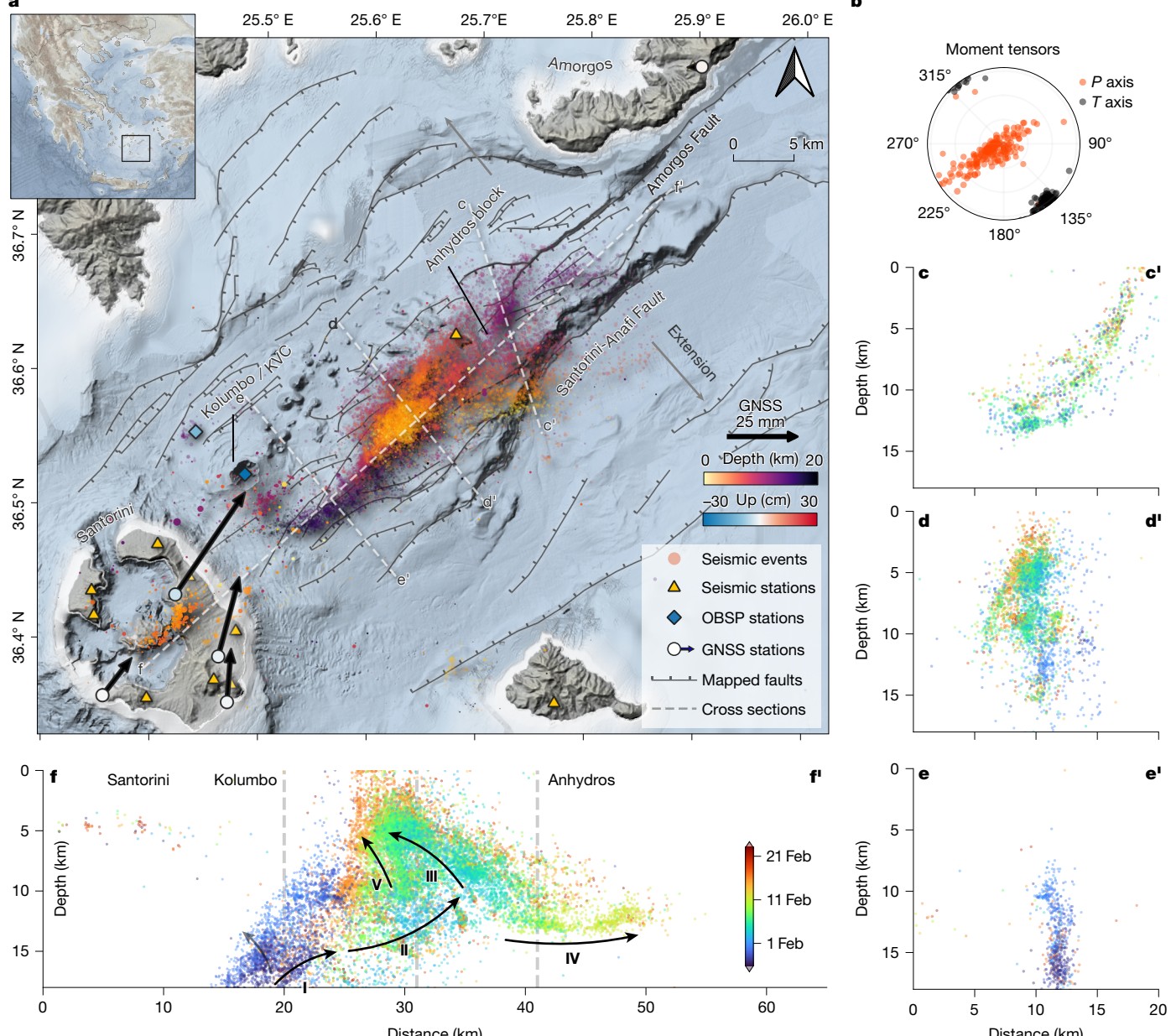

**Fig. 1 | Distribution of seismicity. a**, Map of the seismicity offshore Santorini Island with primary structural features labelled. The high-resolution earthquake catalogue includes more than 30,000 events between 1 October 2024 and 25 February 2025 (Supplementary Video 1). The magnitude of completeness is $M_w$ = 1.3 on average (Extended Data Fig. 2). Seismicity is coloured by depth. Seismometer stations are shown as yellow triangles. Black arrows indicate the horizontal displacement of the continuous GNSS stations on Santorini from 25 January to 25 February 2025. **b**, Moment tensors pressure (*P*) and tension (*T*) axis orientations of 180 events with $M_w$ > 3.6. **c**–**f**, Vertical cross-sections through the seismic swarm colour-coded with time (25 January to 25 February 2025). **f**, Succession of spatiotemporal seismic Phases I to V. Vertical dashed lines show the locations of cross-sections **c**–**e**.

and ground deformation recorded in 2011–2012 indicate the presence of magma bodies at various depths and upwards fluid migration[13–18]. At the same time, Santorini and Kolumbo are part of the highly active Santorini–Amorgos Tectonic Zone, which is characterized by an en-echelon rift system of several grabens separated by relay ramp structures[19,20], which produced the 1956 tsunamigenic $M_w$ = 7.5 Amorgos earthquake doublet[21,22]. Given the area's hazardous history, the origin of the 2025 crisis has been the subject of intense debate to clarify the volcanic or tectonic nature of the earthquake swarm.

Here we provide a comprehensive model of the geological processes during the observed unrest by integrating a high-resolution machine learning earthquake catalogue, generated automatically in near real-time[23], with earthquake source inversions, tremor analysis,

ocean-bottom seismometer and pressure sensor (OBSP) recordings and satellite-derived surface deformation data. We explore the interplay between a dike intrusion, neighbouring volcanoes, crustal magma bodies and faults, which enables the reconstruction of the spatiotemporal evolution of seismic activity and the assessment of dynamic volcanic processes, including the coupling of magmatic feeding systems.

## Spatiotemporal evolution

Before the seismic crisis, from July 2024 to January 2025, a gradual uplift was detected within the Santorini caldera. Approximately 45 mm of absolute displacement was recorded at the Global Navigation Satellite System (GNSS) station SANT on Santorini (Fig. 3b). Also, interferometric

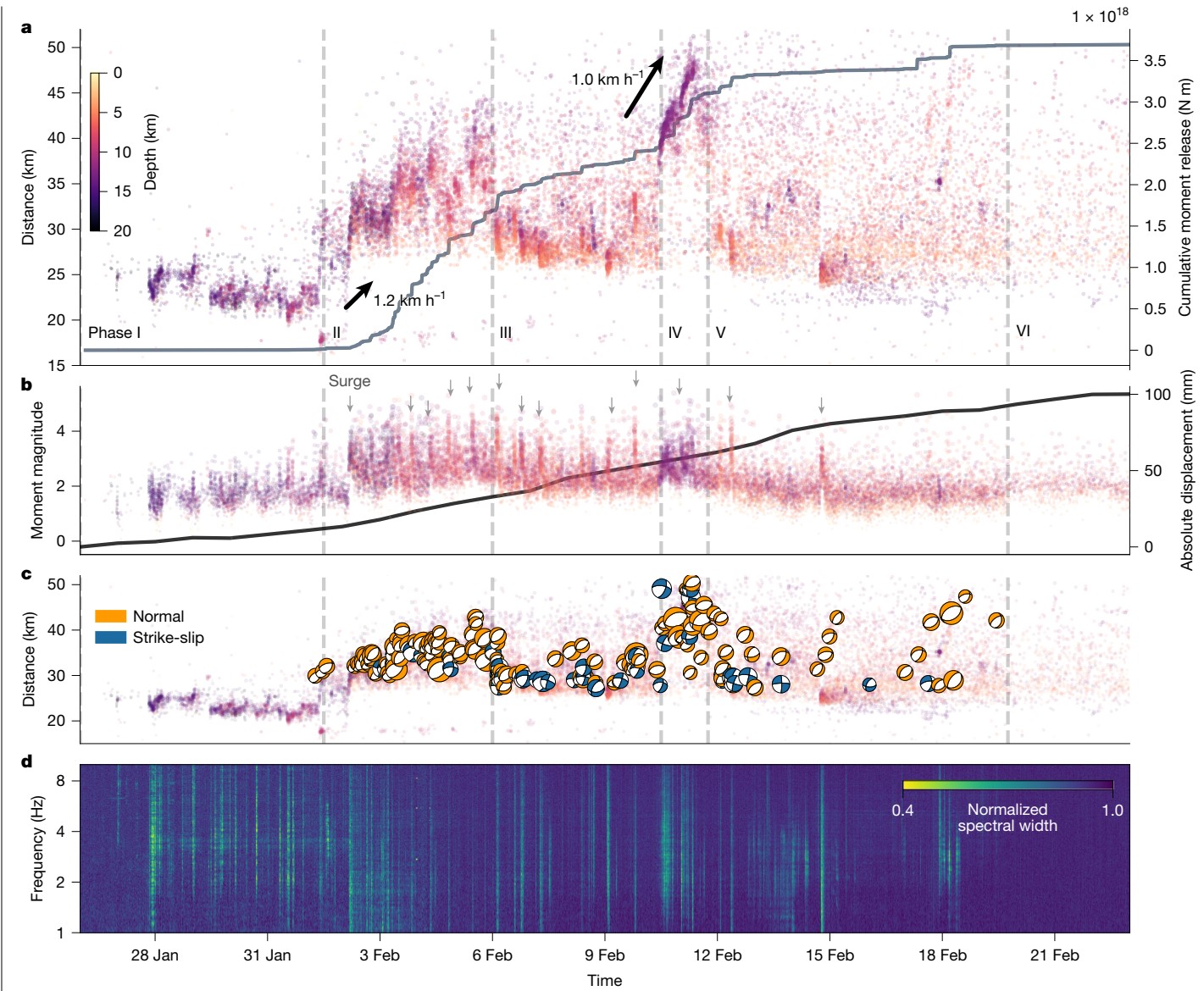

**Fig. 2 | Spatiotemporal evolution of seismicity. a**, Migration of seismicity shown by the hypocentral distance from Santorini island, coloured by depth with annotated apparent dike migration velocities during phase II and notable diking in the northeast, phase IV (Extended Data Fig. 4). A grey line in the background shows the cumulative moment derived from $M_W$. **b**, Moment magnitudes are coloured by depth, and grey arrows indicate intense dike propagation periods. The absolute displacement of the GNSS station SANT on Santorini is shown as the black line in the background. **c**, Seismic moment tensors coloured by focal mechanism and scaled by magnitude. **d**, Coherent seismic tremors detected between 1 and 10 Hz on the vertical channel of the seismic stations within 1° distance of the swarm centre. Low 'spectral width' indicates a highly coherent tremor wavefield in contrast to non-coherent noise.

synthetic aperture radar analysis (InSAR) of Sentinel-1 satellite data showed inflation corresponding to approximately 50 mm deformation in line of sight (LOS). Volcanic unrest continued into January 2025, when low-magnitude seismic activity further intensified around the northern caldera basin of Santorini (phase 0; Fig. 1, Extended Data Fig. 1 and Supplementary Fig. 1), associated with increased $H_2$ and $CO_2$ gas flux at Santorini's central volcanic island of Nea Kameni (Extended Data Fig. 3). On 27 January 2025, 19:00 UTC, the seismicity rate soared abruptly, with a seismic swarm initiating 5 km west of Santorini at depths between 12 km and 18 km with low-magnitude ($M_W$ = 1–3) earthquakes accompanied by strong coherent seismic tremor (Figs. 1 and 2d). During phase I (days 1–5), seismicity initially propagated towards Santorini before migrating northeastwards away from the island. Phase I also marked the onset of rapid surface deformation, characterized by northeastwards subsiding motion measured by GNSS on Santorini and subsidence at OBSP stations at Kolumbo (Fig. 3).

Starting 1 February (phase II; days 6–10), earthquake hypocentres became shallower, migrating northeastwards towards Anhydros at a velocity of approximately 1 km h⁻¹, intensifying at a depth of about 8 km with earthquakes exceeding $M_W$ = 5. From 6 February (phase III; days 11–15), activity increased again towards the southwest, concentrating about 7 km southwest of Anhydros on the uppermost 5 km of the crust. Between 10 and 11 February, during phase IV, deep seismicity (initiating at 12 km depth) migrated 25 km subhorizontally northeastwards across the Anhydros block with peak propagation velocities of about 1.0 km h⁻¹ (Extended Data Fig. 4). During phase V (days 18–25), seismicity returned to the previously active shallow cluster, with several seismic surges concentrated below 5 km. Since 20 February, in phase VI, activity has generally decreased. The surface displacement of Santorini by this time exceeded 10 cm (Station SANT; Extended Data Fig. 5), and subsidence reached 18–32 cm at Kolumbo's crater floor and 6–13 cm on its northern flank (Extended Data Fig. 6).

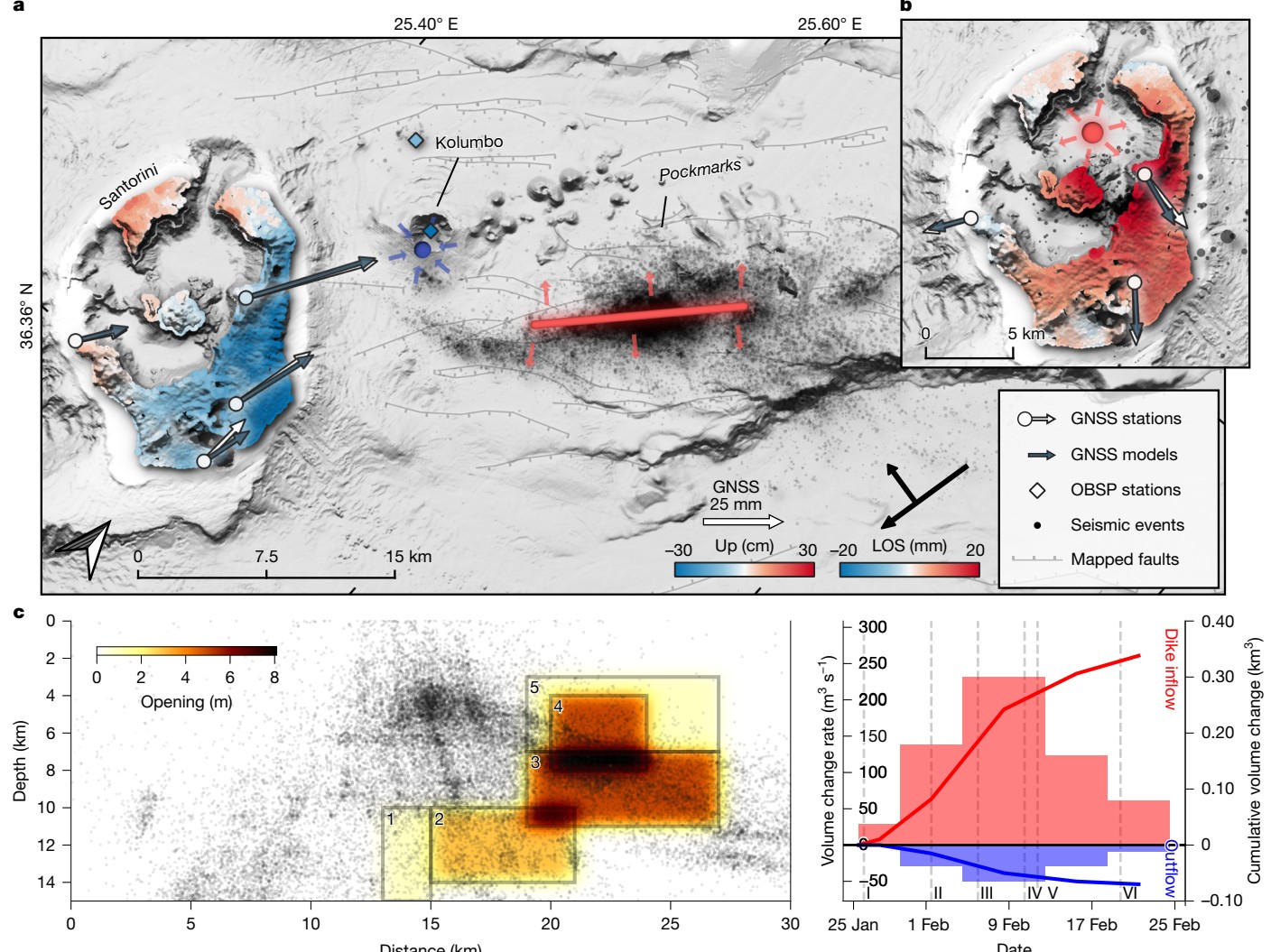

**Fig. 3 | Surface deformation and geodetic modelling. a**, Map showing GNSS, OBSP and InSAR inferred surface displacements during diking between 23 February and 3 March 2025 and inverted source models. The joint geodetic inversion of InSAR, GNSS and OBSP locates the deflating mid-crustal magma reservoir (blue circle) beneath Kolumbo at 7.6 km depth with a 95% confidence interval (Supplementary Figs. 3 and 4) between 6.9–8.2 km and a volume decrease of 0.076 km³ (confidence interval 0.069–0.082 km³). The vertical inflating dike (red line) has a length of approximately 13 km and extends from 5 to 11.5 km in depth, aligning with the seismicity (black dots). The volume increase in the dike is 0.313 km³ (confidence interval 0.296–0.330 km³). **b**, Inflation between July 2024 and January 2025, measured by InSAR and GNSS with a

modelled expanding magma reservoir at 3.8 km depth (confidence interval 2.4–5.4 km) and a volume change of 0.004 km³ (confidence interval 0.002–0.007 km³), averaging an influx rate of 0.26 m³ s⁻¹ (confidence interval 0.13–0.48 m³ s⁻¹). **c**, Time-dependent diking model during diking between 25 January and 24 February 2025 in five consecutive time windows along the inverted dike (**a**), on the basis of joint inversion of InSAR, GNSS and OBSP. The cumulative opening has been smoothed with a 0.2 km Gaussian filter. Seismicity along the profile is plotted as black dots. **d**, Magma flow rates (red and blue bars) of the deflation source beneath Kolumbo (outflow) and dike inflation (inflow) and cumulative volumes (red and blue lines). Vertical dashed lines mark the temporal phases of seismicity (Fig. 2).

We identified at least 12 vigorous seismic surges (grey arrows in Fig. 2b), characterized by a sharp increase in seismicity rate, an increase in magnitude and concurrent tremor activity, followed by a lateral migration of the inferred seismic front (Fig. 2a,b). This dynamic migration pattern, along with the substantial surface deformation observed by GNSS, OBSP and InSAR, is characteristic of magmatic dike intrusions in extensional domains[6,24–27]. Seismic tremor episodes coincided with intense rupture propagation periods (Fig. 2d). The tremor signals contain high-frequency energy (greater than 5 Hz), consistent with their composition of microquakes in highly active swarms. We analysed 180 earthquake moment tensors for events with $M_W > 3.6$ using a Bayesian full-waveform inversion (Methods), showing a consistent NW–SE-oriented tension axis perpendicular to the earthquake migration path, consistent with normal faulting on existing fractures aligning with the seismic swarm's geometry (Figs. 1b and 2c and Extended Data

Figs. 7 and 8). The observed focal mechanisms range from normal faulting to strike-slip and oblique faulting, typical in extensional settings[5,8]. Most events exhibit non-double-couple terms with positive isotropic and compensated linear vector components, indicating complex, dilational processes accompanying shear faulting. Phases of lateral migration (II and IV) are dominated by normal faulting, and episodes of upwards propagation of seismicity (III and the beginning of IV) are characterized by a notable increase of low-magnitude strike-slip events that focus on an area southeast of Anhydros (Fig. 3c and Extended Data Fig. 7). Such spatiotemporal evolution resembles other dike intrusions in extensional settings, including Iceland[25], Kilauea[6] and Miyake-jima[5].

Analysis of our high-resolution earthquake catalogue (Methods; Supplementary Video 1; Supplementary Data), containing more than 30,000 events and 180 moment tensors, shows a segmented, laterally ascending complex dike intrusion characterized by incremental,

non-uniform and rapid expansion. Periods of dike growth coincide with intense seismicity and large-magnitude earthquakes ($M_W > 5$) concentrated at the advancing dike tip[24] (Fig. 2a,b). Between the surges, when the dike boundary is stationary, seismicity was predominantly lower magnitude ($M_W = 1–3$). This is consistent with gradual pressurization due to a continuous inflow of fresh magma until peak stresses at the boundary are large enough to trigger dike growth, accompanied by larger earthquakes[24,28].

## Magma migration induced ground deformation

Analysis of ground deformation before and during diking provides constraints on the spatiotemporal migration of magma. Continuous GNSS and InSAR time series from 10 July 2024 to 18 January 2025 show a concentric gradual uplift of the Santorini caldera of approximately 50 mm during the prediking preparatory phase. Our geodetic inversion results indicate inflation of a mid-crustal magma body in the Santorini caldera at a median depth of 3.8 km (Fig. 3b and Extended Data Fig. 9), corresponding to a volume increase of 0.004 km³ and an inflow rate of 0.26 m³ s⁻¹. The inflation location and depth in 2025 match the source, which inflated during the 2011–2012 unrest[13], indicating a renewed magma or fluid inflow into the same reservoir. The inferred volume change is smaller than the two magma pulses during the 2011–2012 Santorini unrest, which each delivered about 0.01 km³ to the shallow magmatic reservoir beneath the Santorini caldera[13].

After 27 January, with the initiation of the dike intrusion, significant ground deformation was recorded by GNSS stations on Santorini and the surrounding islands and by OBSP at Kolumbo volcano (Fig. 3a). A joint inversion of GNSS, OBSP and InSAR observations indicate two sources: (1) a volume decrease in a mid-crustal reservoir beneath Kolumbo volcano and (2) a uniform opening along an approximately 13-km-long dislocation colocated with seismicity, extending between Kolumbo and Anhydros, representing the emplacement of a dike. The inversion results (Methods) indicate that the deflation source is at a median depth of 7.6 km beneath Kolumbo (5–6 km northwest from the nucleation point of seismicity), with an inferred median volume decrease of 0.076 km³. The median volume increase in the dike is 0.313 km³ (Fig. 3a and Extended Data Fig. 10), which is approximately four times greater than the modelled volume loss of the mid-crustal magma reservoir.

By splitting the continuous deformation time series into five time intervals, we produce a time-dependent model of the inflating dike intrusion and concurrently deflating mid-crustal reservoir (Fig. 3c,d and Supplementary Fig. 2). After dike initiation, the inflow rate increased rapidly and peaked between 4 and 12 February (phases II–IV), with average inflow rate exceeding 200 m³ s⁻¹. Subsequently, the inflow rate decreased to almost zero on 24 February, probably reflecting the end of this intrusion.

## Dynamics, interactions and coupling

Although the last eruption of Santorini dates back to 1950, three pulses of magma recharge into its shallow magma reservoir at approximately 4 km depth since 2011 have shown moderate inflow rates of 0.3–0.6 m³ s⁻¹. The cumulative inflow from these pulses is approximately 0.024 km³, which neither resulted in diking at Santorini[13,29] nor triggered activity at the neighbouring Kolumbo volcano. Similarly, two earthquake swarms at Kolumbo in 2003 and 2006, interpreted as episodes of fluid or magma ascent towards its shallow magma reservoir at 2.5 km depth, also did not affect the Santorini system[14,19]. Although tomographic studies remain inconclusive about a connection between Kolumbo's and Santorini's magmatic plumbing system[30,31], petrological evidence indicates distinct mantle sources and magma differentiation in their respective shallow reservoirs[3,4]. However, during the 1650 Kolumbo eruption, simultaneous ash or gas emissions from the Kameni

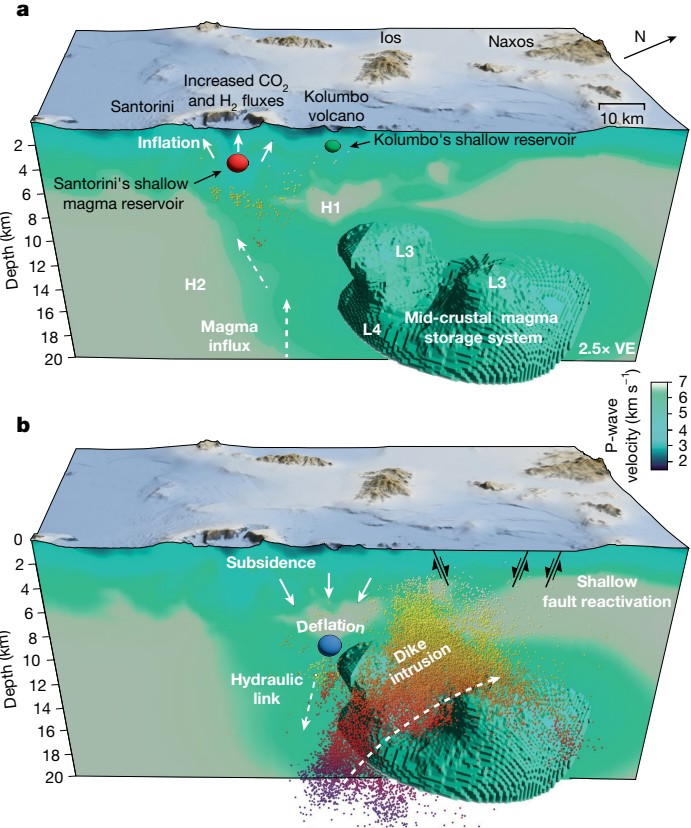

**Fig. 4 | Evolution of the 2025 Santorini crisis. a**, The preparatory phase from June 2024 to January 2025. The influx of magma from a mid-crustal magma reservoir into the shallow magma reservoir beneath Santorini resulted in surface uplift and increased seismicity at the caldera. The green 3D body shows the locations of a low P-wave velocity anomaly L4 (21% lower $v_P$). High-velocity anomalies H1 and H2 are interpreted as rheologically strong layers, whereas low-velocity anomalies L3 and L4 represent branches of a mid-crustal magma storage system at different depths (see refs. 30,31). **b**, Volcano–tectonic crisis: diking and faulting. Starting on 27 January 2025, a dike intrusion was initiated from a mid-crustal magma reservoir and propagated laterally upwards into the Anhydros block. Seismicity (coloured dots) occurred at the tip and boundaries of the dike rupture, and shallow crustal normal faults were activated by changes in stress and fluid pressure (black lines). The concurrent deflation of a shallower mid-crustal melt reservoir led to subsidence, which was measured at Santorini and Kolumbo. Scale bar, 10 km. VE, vertical exaggeration.

islands are reported by one historical source[32]. The 2025 Santorini crisis offers an unprecedented opportunity to study the reactivated volcanic systems through independent observations of geodetic deformation and seismicity. The sequential timing of shallow magma chamber inflation at Santorini starting in July 2024, followed by the dike intrusion into the Anhydros Block, with concurrent deflation at Kolumbo, allows for the reconstruction of magma migration pathways and points to coupling between the reservoirs over tens of kilometres (Fig. 4).

The cascading course of events during the crisis may indicate that both systems share and possibly compete for the same magma supply at depth, implying a hydraulic connection between the neighbouring volcanoes[12,33], which explains the consecutive reservoir inflation at Santorini and deflation at Kolumbo during the crisis. Whether the deeper reservoir beneath Kolumbo was also supplied during the inflation period of Santorini is unclear. Reservoirs beneath both volcanoes may interact through crustal stress transfer or pressure changes in the fluid plumbing system at depth[12,34–36], although there is no evidence for a direct connection between the shallow magma reservoirs. Similar mechanisms have been proposed for intrusions and eruptions in

Hawaii, Iceland and Kamchatka[37–39]. In such coupled systems, the integrity of each magma reservoir controls diking or eruptive activity, with rupturing occurring at the volcano whose reservoir is closer to failure under given inflow rates[34].

The 2025 Santorini–Kolumbo dike intrusion nucleated in the direct vicinity of a low P-wave velocity ($v_P$) anomaly L4 (Fig. 4) previously identified through seismic tomographies[30,31]. Low-$v_P$ anomalies L3 and L4 extend from east of Santorini and south Kolumbo upwards to the NE beneath the Anhydros block and represent a branching lower- to mid-crustal magma storage system containing at least 5–14% partial melt[30,31]. On the basis of our high-resolution seismicity catalogue, the dike intrusion initiated at approximately 18 km depth, whereas deeper magma migration was probably aseismic[40]. First, the dike ascended through the low-$v_P$ anomaly L4 (phase I). Then it migrated shallower and to the northeast, wrapping around storage zone L3 and focusing above L3 at approximately 5 km depth (phases II–V), which marks the phase of highest inflow rates (Fig. 3c). The subhorizontal dike migration during phase IV lies within the deepest and northeastern-most extent of L3. The modelled location of codiking magma depletion at $7.5 \pm 1$ km depth beneath Kolumbo correlates with both (1) a concurrently occurring cluster of vertically oriented seismicity at 8.5–13 km depth and (2) the base of the high-$v_P$/low-$v_S$ (S-wave velocity) anomaly H1, previously interpreted as a rheologically strong layer with vertical, melt-filled cracks[14,30]. These combined observations indicate that magma is transferred vertically between separate branches of the mid-crustal storage system, implying a vertical hydraulic linkage between the deflation source and the dike-feeding system[12,41].

The spatiotemporal evolution of seismicity indicates the creation of magma pathways above the mid-crustal low-velocity bodies L3 and L4 (Figs. 1 and 4), associated with several bursts of accelerated dike propagation into the Anhydros block (Figs. 2 and 4). The dike intrusion caused local and regional poroelastic stress changes because of opening and hydrothermal activation, inducing and triggering seismicity[42] in the highly faulted basement of the Anhydros block. The extent to which the normal-faulting earthquakes have released tectonic prestress because of the dike's emplacement remains unknown. The Anhydros block is a tilted block that probably represents a large relay ramp structure between the bounding Amorgos and Santorini–Anafi faults[20]. Distributed faulting and fracturing in relay ramps create broad damage zones, providing preferential pathways for vertical fluid migration[43]. The regional tectonic stress field and reactivation of structural elements primarily controlled the northeasterly strike direction of the dike. Fault traces on the seafloor and seismic reflection data indicate that the shallow subsurface is disrupted by various sub-kilometre-scale fault zones that control the location of fluid escape structures on the seafloor, evidenced by strings of pockmarks and the linear alignment of volcanic centres of the Kolumbo volcanic chain (KVC), both of which form atop southeast-dipping normal faults (Fig. 1a and Extended Data Fig. 11).

The hazard of future volcanic eruptions depends on the volume and composition of magma recharge from depth, possible mixing of different magmas, the location of ascent and water depth at the point of breach. The modelled ratio between dike volume and reservoir drainage during the 2025 diking event is approximately 4, which is significantly lower than observed for evolved gas-rich systems such as the shallow reservoir at Etna or Dallol[44,45], with ratios of 17 and 32, but slightly higher than observed for primitive basaltic systems such as Hawaii or Iceland[46], indicating a rather deep, primitive magma source. The mid-crustal magma reservoirs of Santorini and Kolumbo are thought to contain mafic to evolved, intermediate magmas that feed low-fraction silicic melts into shallow crustal reservoirs beneath Kolumbo and Santorini[3,4]. The highly explosive 1650 eruption of Kolumbo involved the injection of mafic magmas into the shallow felsic reservoir[47], indicating that the interaction between different reservoirs and the mixing of magma batches play an essential role during eruptions of the Kolumbo system.

Most cones of the KVC are monogenetic eruptive centres, generally associated with the rapid ascent of small batches of relatively primitive magma. Bulk volumes of less than 0.2 km³ dense-rock equivalent[48] (assuming 75% deposit porosity[1,2]) indicate that they were formed by pulses of magma with volumes similar to the 2025 dike intrusion. The cones' lack of pronounced summit craters indicates a mildly explosive formation, and the absence of slope failure scars indicates minor tsunamigenic potential[49]. Water depths are an important control of the explosivity of submarine eruptions. Although the KVC cones have formed in water depths greater than 400 m, probably suppressing phreatomagmatic explosions[50], the 2025 dike intrusion was located in shallow water depths (approximately 200 m). Thus, the magma ascent path through the shallowest crust will strongly influence the hazard potential of future eruptions, highlighting the importance of better understanding the local volcano–tectonic framework.

Our study demonstrates the complex interplay between coupled volcanic plumbing systems. Although there were clear precursory signals to the crisis, including significant inflation in the Santorini caldera, elevated seismic activity and increased gas fluxes, it would have been impossible to predict the cascading evolution culminating in the 2025 dike intrusion. This emphasizes the importance of integrated high-resolution monitoring of seismicity, surface deformation and hydrothermal activity in real time, facilitating improved hazard assessment and early warning.

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

## Methods

### High-resolution earthquake catalogue

The National Observatory of Athens Seismic Network (code HL) and the Aristotle University of Thessaloniki Seismological Network (code HT) recorded seismic data, with stations positioned across Santorini and nearby islands. The two OBSPs (Fig. 1) were deployed on 1 January and recovered on 19 March 2025 and recorded seismic data during the unrest (see 'Marine seismic and bathymetric data'). GFZ deployed three more real-time stations on Santorini in early February 2025. Earthquake events were detected and located in near real time with high resolution using the Qseek stacking and migration framework[23]. The method is based on the neural network PhaseNet trained on Southern California data (SeisBench model original) to identify and annotate P- and S-phase first arrival certainties in continuous seismic waveform data[51,52]. These arrival annotations are back-projected into an adaptive octree grid. The phase arrival signature is stacked to a semblance based on modelled travel times from a local one-dimensional layered seismic velocity model[53] (Supplementary Fig. 5). After initial detection and localization, statistical source-specific station terms (SSST) are extracted[23,54]. Relocating seismicity with SSST corrections can compensate for three-dimensional (3D) seismic velocity heterogeneities, resulting in improved stacking, enhanced event detection and enhanced hypercentre accuracy.

This data-driven method for detecting and simultaneously localizing seismic events propagates phase arrival confidence values to the back-projection stack and into the hypocentre location. With meaningful SSST corrections, this method enables the localization of seismicity and microseismicity in near real time and with high spatial accuracy. The earthquake moment magnitudes were calculated from measured peak amplitudes using forward-modelled attenuation curves[47], derived from the same velocity model used for localization[48].

### Moment tensor inversion

On the basis of the regional broadband seismic data with epicentral distances of up to 700 km from the seismic networks GE, MN, HL, HT and HA (Supplementary Fig. 6), we inverted for probabilistic moment tensors using the Pyrocko Grond framework[55,56]. The inversion was set up to model vertical, radial and transversal full-waveform displacement seismograms in the time domain, applying a bandpass filter in the frequency band 0.03–0.08 Hz and assuming a global velocity model AK135 (ref. 57). All data were visually inspected, and traces showing noisy, tilted and/or saturated records were removed. We inverted for both full and deviatoric moment tensors. The deviatoric moment tensors are presented in Fig. 2c and Extended Data Fig. 6, and the normal and strike-slip events are clustered by the mechanism's rake angle ($\gamma$): normal $-135° \leq \gamma \leq -45°$; strike-slip $-45° \leq \gamma \leq 45°$ (and vice versa, $-180° \leq \gamma \leq -135°$). The full moment tensor inversion results are shown in Extended Data Fig. 7.

The GEOFON moment tensor analysis was performed in the time domain using full waveforms, body waves, surface waves, W-Phase[58] or a combination of these, depending on the earthquake's magnitude and depth. All available data were used in the analysis. Large earthquakes are generally recorded with a higher signal-to-noise ratio and are typically analysed using more seismic stations compared to minor earthquakes. In the context of the events examined in this paper, however, we maximized the consistency of the data used: to locate the hypocentres, we used only the nearest seven stations (GE.THERA, GE.APE, GE.KARP, GE.ZKR, GE.IMMV, HL.ARG, GE.KTHA), which are favourably located around the target region. Point-source, deviatoric seismic moment tensors were inverted using full waveforms from stations within 1,000 km of the epicentre, bandpass filtered between 30–80 s throughout the analysis. A depth search was performed, keeping the independently determined epicentre fixed. The Earth model used was PREM[59].

### Seismic tremor detection

We detected tremor activity using the covseisnet method[60], which was initially developed to study volcanic tremor[61]. This approach identifies coherent signals across a seismic network by quantifying wavefield coherence through the spectral width of the covariance matrix. We analysed continuous seismic data from vertical broadband (channel code HHZ) components within a 1° radius of the swarm centre. Waveforms were downsampled to 50 Hz, detrended and bandpass filtered between 1 and 10 Hz to enhance tremor-related signals while mitigating microseismic noise[62]. We computed the array covariance matrix $C(t,f)$ in the time–frequency domain[63]. The time signals at each station were divided into 1,000-s-long overlapping averaging windows. Each window $t$ was divided into subwindows of 50 s length with 50% overlap. In each subwindow $m$, the signals were normalized with spectral whitening to suppress short impulsive earthquake signals and to improve coherent tremor detection.

We then computed Fourier spectra $u_n(m,f)$ where $u$ is the Fourier spectrum and $f$ is the frequency for all seismic stations $n = 1, \ldots, N$, resulting in a vector:

$$\mathbf{U}_m(t,f) = [u_1(t_m,f), u_2(t_m,f), \ldots, u_N(t_m,f)]^T$$

The covariance matrix in window $t$ was computed as follows:

$$C(t,f) = \frac{1}{M} \sum_{m=1}^{M} \mathbf{U}_m \mathbf{U}_m^\star$$

where $*$ denotes the hermitian transpose. The coherence of the wavefield across all stations is defined as the spectral width, $\sigma(t,f)$:

$$\sigma(t,f) = \frac{\sum_{i=1}^{N} (i-1)\lambda_i(t,f)}{\sum_{i=1}^{N} \lambda_i(t,f)}$$

Here $\lambda_i(t,f)$ are the $N$ eigenvalues of the covariance matrix $C(t,f)$, which are sorted in decreasing order. Lower spectral width values correspond to a stronger network coherence with a single dominant source, whereas an incoherent wavefield produces higher spectral width values.

### GNSS surface displacement

Several permanent GNSS stations have been installed on Santorini and the wider central Aegean region (Fig. 1). Since 2006, the SNTR station has been operational. The Hellenic Cadastre company has installed five stations in the area, one in Santorini (048A) and the others in the surrounding region (047A, 049A, 050A, 097A). The station SANT has been installed in collaboration with the University of Athens and METRICA, and station SNTJ has been installed and maintained by JGC. Seven stations were available for the initial inflation period from August 2024 to February 2025. There are also four other stations in the area for which data are not freely available. Since early February, because of the continuing crisis, various research teams have installed at least 12 more stations at new locations to monitor ground displacements.

GNSS data were analysed in an automated scheme daily using the Bernese GNSS software (v.5.2)[64]. Both GPS and GLONASS observations were used when available. The reference frame used was IGS20, aligned using a set of 19 IGS stations, following the IERS Conventions 2010. The processing followed EUREF standards outlined in the Guidelines for Analysis Centres. IGS and CODE products were applied in the processing, and ocean loading corrections were implemented using the FES2004 model. A 3° elevation cut-off angle was set, with elevation-dependent weighting applied. GMF and/or VMF1 were used for tropospheric modelling, incorporating the Chen–Herring gradient parameter. Ambiguities were resolved using a length-dependent algorithm. GLONASS observations were included when available.

This approach ensures consistency and reliability in the GNSS data processing, aligning with established geodetic standards. Following the estimation of daily site coordinates, position time series were stacked for every network site. The latter were, in turn, analysed using the Hector software[65].

## InSAR surface deformation

To monitor the evolution of ground displacements over Santorini and its neighbouring islands, we used multitemporal InSAR using data from the Sentinel-1 SAR mission, which provides a nominal spatial resolution of 5 m × 20 m and a temporal resolution of 12 days. Our analysis was based on three Sentinel-1 tracks covering Santorini and neighbouring islands from March 2024 to March 2025: ascending track 029 (64 images), descending track 036 (66 images) and descending track 109 (64 images). All SAR images were coregistered for each track and resampled to a common reference image using GAMMA software. Then, we estimated the flat-earth component and the topographic phase contribution using the GLO-90 Copernicus digital elevation model (DEM; European Space Agency) and subtracted them from the resampled SLC stack before time-series analysis. We performed the time-series analysis in SARvey software[66]. For each track, we built a small baseline network of interferograms with dense connections between each image and ten other images in the stack while maintaining a maximum temporal baseline of 132 days. The interferograms were multilooked by a factor of 10 in range and 2 in azimuth direction to enhance the signal-to-noise ratio. Next, pixel selection was based on the temporal phase coherence threshold of 0.9, estimated over a 9-by-9 window across all interferograms[67]. Phase unwrapping was carried out on the selected pixels using the Phase Unwrapping Max-Flow method[68]. After phase unwrapping, we inverted the network of interferograms by means of least squares to estimate the time series of deformation. The time-series processing was performed in radar coordinates before the results were ultimately geocoded into the WGS84 coordinate system.

## Geodetic modelling

Geodetic inversions using a combination of GNSS, InSAR and OBSP observations were undertaken using a modified version of the GBIS software[69] to determine the likely deformation sources responsible for the observed displacements. This Bayesian inversion software uses a probabilistic Markov-chain Monte Carlo approach to generate probability distribution functions for each modelled parameter, providing a median solution and model confidence intervals. During the initial prediking inflation period, we used LOS displacements from three Sentinel-1 tracks as input to the inversion, spanning the period from 10 July 2024 to 18 January 2025, from one ascending and two descending tracks (029, 036 and 109). The inflation observed during this period can be explained by the pressure increase in a single spherical-type source modelled using a Mogi point source[70]. For the diking period (28 January to 23 February 2025), we also used LOS displacements from the same three Sentinel-1 tracks above (spanning 18 January to 23 February 2025), in addition to GNSS displacements from eight stations and vertical displacements from two OBSP stations. The deformation that occurred during this latter period can be explained by concurrent pressure decrease in a spherical-type source (close to Kolumbo) and inflation of a dike, modelled using a deflating Mogi point source and uniform opening along a subvertical Okada dislocation[71], respectively. Only the dip and the centre point of the dislocation (dike) were fixed; all other source parameters were solved in the inversion.

We investigated the evolution of the dike over time using GNSS displacements and OBSP depth differences in five consecutive time intervals, ensuring robust data coverage. Satellite InSAR data lack sufficient temporal resolution and were excluded from this analysis. These intervals were selected to cover the entire diking event while maximizing the number of GNSS observations to show significant displacements. For each time interval, we inverted for volume change in a deflating magma body, modelled as a Mogi source[70] and opening along a subvertical dike, modelled as an Okada dislocation[71]. The Mogi source location was fixed (Fig. 3) on the basis of the results of the joint InSAR–GNSS–OBSP inversion detailed above. The inclusion of the GNSS station on Anhydros Island significantly improves the azimuthal data coverage and yields better fits using a 35° strike of the Okada dislocation source. This is well aligned with the seismicity. However, the dike could freely move along a fixed plane (Fig. 3). The inversion was conducted as a grid search, minimizing chi-squared, considering GNSS and OBSP deformation measurements to be relative and not absolute to account for potential network bias. The nature of the applied grid search inversion does not derive error margins or confidence intervals. Thus, we used the static inversion to derive the absolute deflation and intrusion values. However, both methods resulted in very similar numbers. The final model shows a deflating deep magma body and an inflating, propagating dike (Fig. 3). All geodetic models presented here consider deformation sources embedded in a uniform elastic half-space.

## Marine seismic and bathymetric data

Most of the marine geophysical datasets used in this study were collected aboard RV *Maria S. Merian* between December 2024 and January 2025 (Cruise MSM132). High-resolution two-dimensional (2D) and 3D seismic data were acquired using two generator–injector guns in harmonic mode as the seismic source. Two-dimensional seismic profiles were acquired using Geometrics GeoEel streamers with active lengths ranging from 75 m to 125 m. Processing included the application of crooked-line marine geometry, bandpass filtering, sorting to common midpoints, normal moveout correction and post-stack 2D Stolt migration using water velocity (1,500 m s$^{-1}$). The 3D seismic data were acquired across the Amorgos Fault using a P-Cable system with 16 streamers (8 channels each, 1.5625 m apart) attached to a cross cable of approximately 200 m width. Processing included geometry definition, trace editing (removal of dead and very noisy traces), trace balancing, frequency filtering, *f-x* deconvolution and stacking. The stacked data were then sorted into crosslines and migrated with a constant-speed Stolt migration (1,525 m s$^{-1}$) and sorted back into inlines, and finally the same 2D Stolt migration was applied. The seismic data have a horizontal resolution of approximately 6.5 m and a vertical resolution of approximately 5 m in the shallow seafloor (assuming a seismic velocity of 1,600 m s$^{-1}$, a dominant frequency of 100 Hz and a resolution criterion of $\lambda/2$). A combined digital elevation model was constructed using satellite-derived Advanced Spaceborne Thermal Emission and Reflection Radiometer data, a community DEM from the European Marine Observation and Data Network and data collected during the GEOWARN project onboard RV *AEGAEO*, the PROTEUS project onboard RV *Marcus G. Langseth* and the MSM132 research cruise onboard RV *Maria S. Merian*.

## OBSP sensor experiment

During the MSM132 experiments, eight OBSP sensor stations were deployed on 1 January 2025. These consisted of Long-term Ocean Bottom Seismometer for Tsunami and Earthquake Research frames, which were equipped with a three-component short-period (4.5 Hz) seismometer, HTI-04 and HTI-90 hydrophones, Paroscientific Series 8000 absolute depth sensors with digiquartz transducers (0–700 m H2O and 0–1,400 m H2O versions) and RBR solo3 temperature and duet3 temperature–pressure sensors. Two of these instruments (MSM132-OBS3 and MSM132-OBS6) were recovered by RRS *Discovery* on 19 March 2025. The seismic recordings from OBS6 were used for the high-resolution earthquake catalogue, and the pressure sensor recordings were analysed to provide input for geodetic modelling. The pressure sensor data processing included removing tidal signals using a global tidal model and subtracting oceanographic signals using a tide gauge on the island of Kos. Both instruments were recording for a few weeks at the beginning of the crisis, so both sensors are expected to be affected by instrument drift. We have reduced the influence of this drift by calculating

the linear trend before the diking period (5–25 January 2025) and subtracting this from the time series (Extended Data Fig. 6), resulting in a subsidence of 32 cm for OBSP3 and 12 cm for OBSP6 during the diking period. We also calculated the pressure difference of both instruments (after tidal, oceanographic and linear drift correction), resulting in a differential subsidence of 19 cm. We also estimated the linear drift for the entire time series and corrected the data to provide a minimal estimate of vertical motion. This gives a differential subsidence of 12 cm and subsidence values of 18 cm for OBSP3 and 6 cm for OBSP6. These calculations define the ranges of values used in the geodetic modelling.

## Surface gas flux measurement

Surface gas fluxes at Nea Kameni were measured in July 2024 and February 2025. We used a Dräger X-am 8000 portable multigas instrument, connected to a 10.3-cm-diameter and 16.5-cm-long accumulation chamber, to simultaneously measure $CO_2$, $H_2S$, $SO_2$, $H_2$, $O_2$ and $CH_4$. Only the most abundant gases, $CO_2$ and $H_2$, are considered in this study. Field measurements were made under similar environmental conditions: The surface was cleaned for the measurement, and gravel was removed to provide a flat contact surface. The measurement was started to record the atmospheric background signal. The accumulation chamber was placed on the ground and sealed at the bottom with fine-grained material, increasing gas concentrations. After an average measurement duration of two minutes, the chamber was removed from the ground and flooded with fresh air. In a post-processing step, we selected the part of the graph where a continuously increasing gas concentration was observed and calculated the slope using linear regression, yielding the gas flux in ppmv s$^{-1}$.

## Data availability

The seismic waveform data analysed is available from the EIDA node (https://eida.gein.noa.gr/) using the network codes HL[72], HT[73] and HA[74]. The GNSS displacement data are available at http://dionysos.survey.ntua.gr/dsoportal/. SAR data acquired by the Sentinel-1 satellite are freely available at the Copernicus Open Access Hub (https://browser.dataspace.copernicus.eu/). Earthquake catalogues and moment tensor solutions are available in the Supplementary Data.

## Code availability

The package for detecting and localizing earthquakes, Qseek, is openly available at https://github.com/pyrocko/qseek (ref. 23). The package grond for full-waveform moment tensor inversion is available at https://git.pyrocko.org/pyrocko/grond (ref. 55). The covseisnet package code for the seismic tremor analysis is available at https://covseisnet.gricad-pages.univ-grenoble-alpes.fr/covseisnet (ref. 60). The package SARvey for InSAR time-series deformation analysis is available at https://github.com/luhipi/sarvey (ref. 66).

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

**Acknowledgements** We thank I. Yeo, G. Bayrakci, M. Wollatz-Vogt, S. Hölz and H. Zimmer for their support in collecting the OBSP. We are grateful to the captains and crews of the relevant cruises aboard RV *Maria S. Merian* and RRS *Discovery*. We thank the GNSS data providers NOA, NKUA, the University of Patras, Hellenic Cadastre, Metrica and JGC. These institutes and private companies provide RINEX data of permanent stations free of charge. Maps and spatial analysis were created using QGIS. Europe terrain data were produced using Copernicus data and information funded by the European Union – EU-DEM layers. This study was supported by grant no. FKZ: 03F0952C of the German Federal Ministry of Research, Technology and Space (BMFTR) as part of the DAM mission mareXtreme, under project MULTI-MAREX. T.R.W. and D.M. are supported by ROTTnROCK, a research project funded by the European Research Council, under the European Union's Horizon Europe Programme (grant no. ERC-2023-SyG 101118491). M.M.P. and V.D. are supported by the ISVOLC project funded by the Icelandic Research Fund.

**Author contributions** M.I. and J.K. led the research and conceived the main manuscript with input from P.N., M.M.P., C.B., H.K., T.D., T.R.W. and E.R. M.I. generated the earthquake catalogue, developed the signal-processing method and produced Figs. 1–3. J.K. produced Fig. 4. S.C. and J. Saul analysed the earthquake moment tensors. Q.H., J. Soubestre, N.M.S., F.B. and J.M. analysed the seismic tremor. S.H. analysed the earthquake magnitude statistics. D.A., M.T. and K.R. acquired and processed the GNSS deformation data. M.H.H. and M.M. processed the InSAR data. M.M.P. and V.D. performed the geodetic modelling. E.R., M.I., T.D. and T.R.W. conceived the dynamic diking model. J.K., C.B., G.J.C., M.B.J., J.P., C.H., D.L. and M.U. acquired, processed and interpreted the marine geophysical data. G.J.C., C.B., J.K., J.P., C.H. and M.U. developed the geologic structural model. E.E.E.H., R.S.H. and K.R.A. acquired marine geophysical data and contributed to the 3D tomographic velocity model and interpretation. D.M. measured and analysed the surface gas flux at Nea Kameni volcano. All authors contributed to editing and revising the manuscript text.

**Funding** Open access funding provided by GFZ Helmholtz-Zentrum für Geoforschung.

**Competing interests** The authors declare no competing interests.

**Additional information**
**Correspondence and requests for materials** should be addressed to Marius P. Isken or Jens Karstens.

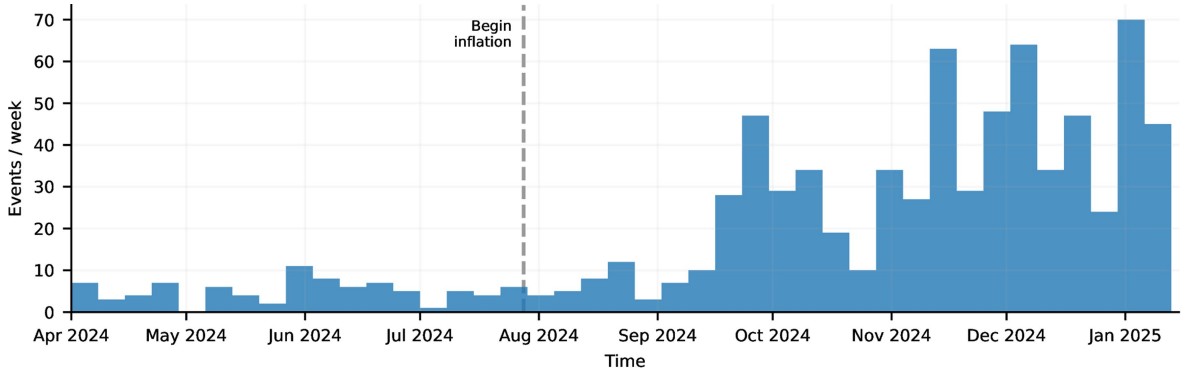

**Extended Data Fig. 1 | Temporal seismicity analysis.** Weekly seismicity rates from April 2024 to 15 January 2025 show significantly elevated earthquake activity beginning in September 2024, following the onset of surface inflation on 28 July 2024.

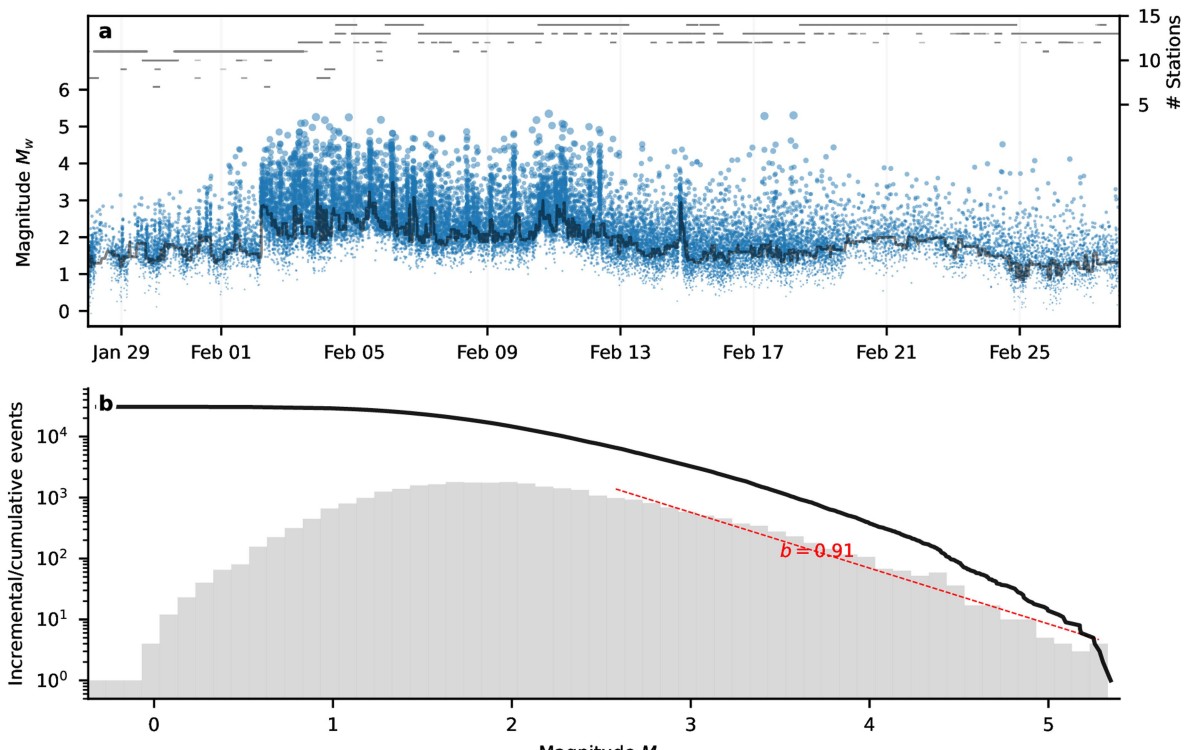

**Extended Data Fig. 2 | Analysis of earthquake magnitudes. a**, Moment magnitudes of detected seismicity as a function of time, where the black line refers to the magnitude of completeness calculated using the maximum curvature method (MAXC[75]) for 100 consecutive events. The number of seismic stations used for each earthquake hypocentre location is shown at the top with the scale on the right. **b**, The corresponding frequency-magnitude distribution, with the estimated Gutenberg-Richter b-value for $M_w > 3$.

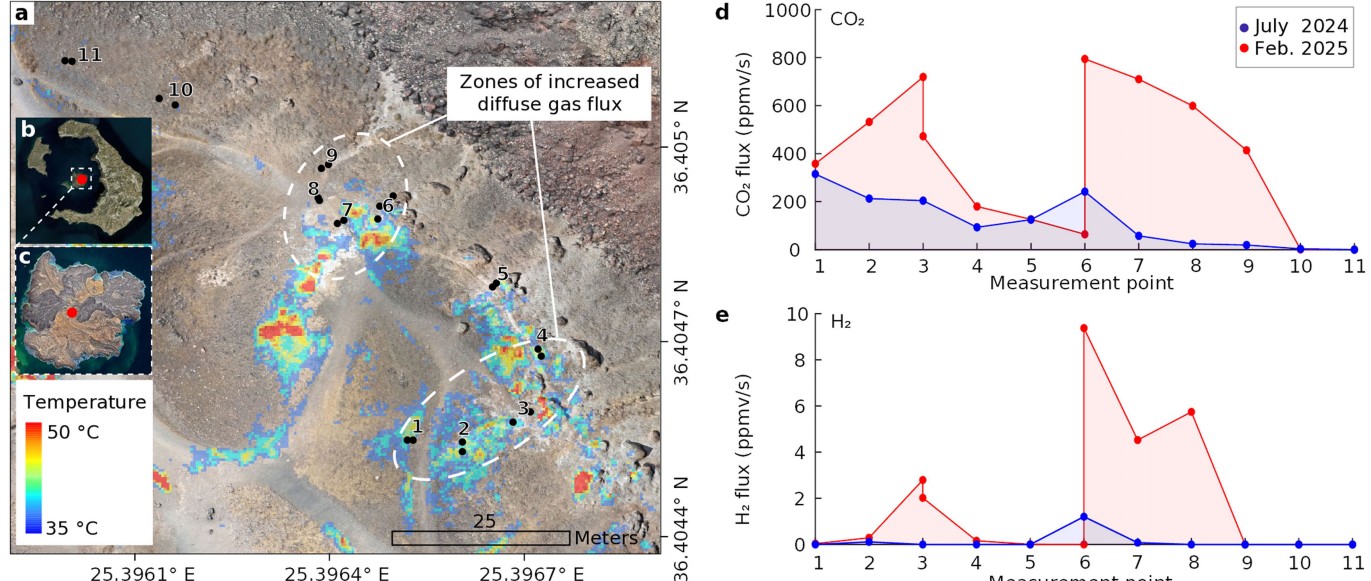

**Extended Data Fig. 3 | Evolution of surface gas emissions at Santorini.**
**a**, Surface gas flux measurements inside the Santorini caldera at Nea Kameni were carried out before and during the magmatic unrest, i.e. on 8 July 2024 and 3 February 2025 at 11 measurement points (marked by black dots) at the central craters of Nea Kameni (location in panels **b** and **c**), using a portable multi-gas instrument connected to an accumulation chamber. Hydrothermal activity and significant surface gas flux occur in several zones with higher surface temperatures and bleached surfaces. We measured in these zones (MP 1–4 and 6–9) and the periphery (MP 5, 10, 11). **c**,**d**, The $CO_2$ and $H_2$ fluxes have increased significantly in these zones from July 2024 to February 2025. This could indicate a general increase in hydrothermal activity due to the recharge of the magma chamber beneath Santorini. However, no significant changes in surface temperatures have been observed so far. But this seems plausible as the surrounding water strongly buffers the system. Measurement points 5, 10, and 11 don't show significant changes.

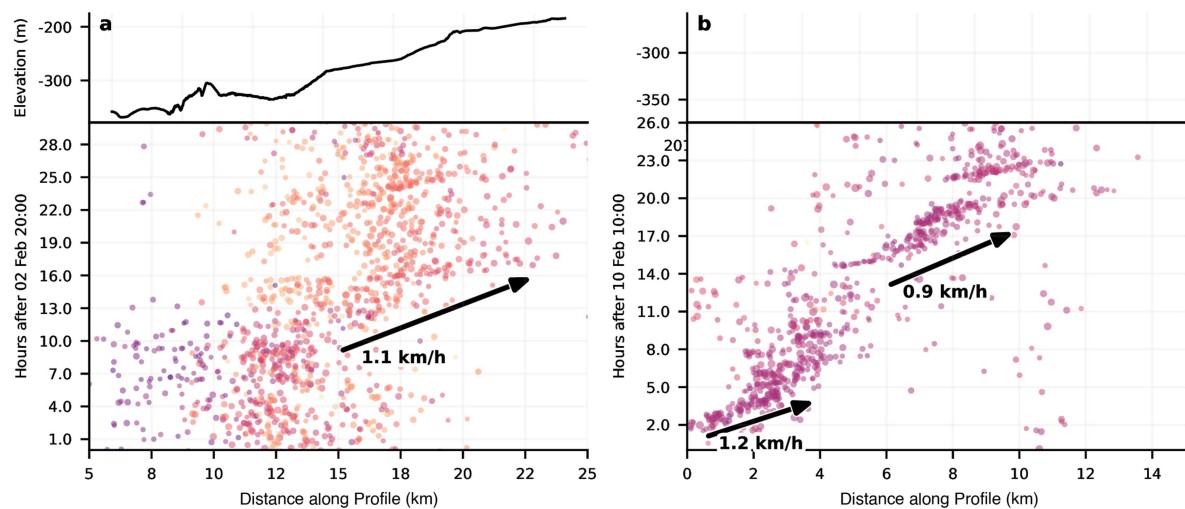

**Extended Data Fig. 4 | Analysis of dike propagation dynamics. a**, Dike propagation velocities and bathymetry (top panels) during the initial Phase II along profile f (Fig. 1f). **b**, Focuses on the rapid deep dike expansion towards the NE during Phase IV.

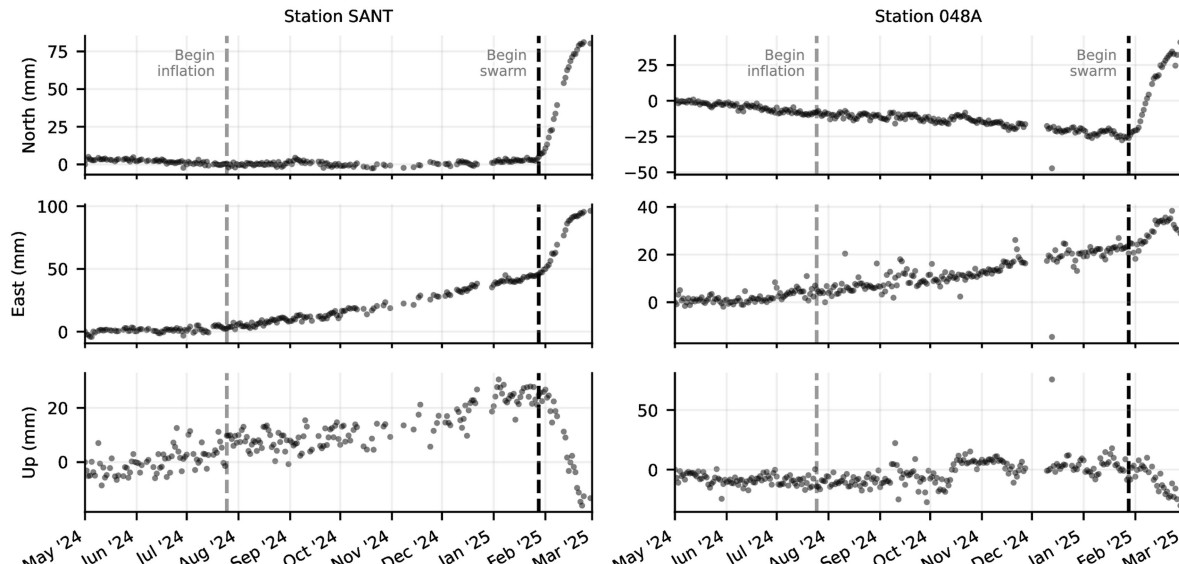

**Extended Data Fig. 5 | GNSS surface displacements on Santorini.** Surface displacements of Santorini island measured by GNSS stations SANT and 048 A between 1 April 2024 and 28 February 2025. Both stations display gradual displacement beginning on 28 July 2024. Until January 2025, SANT was displaced 50 mm. With the onset of the seismic swarm on 25 January 2025, the stations began to show subsidence, with absolute displacements of 96 mm (SANT) and 58 mm (048 A).

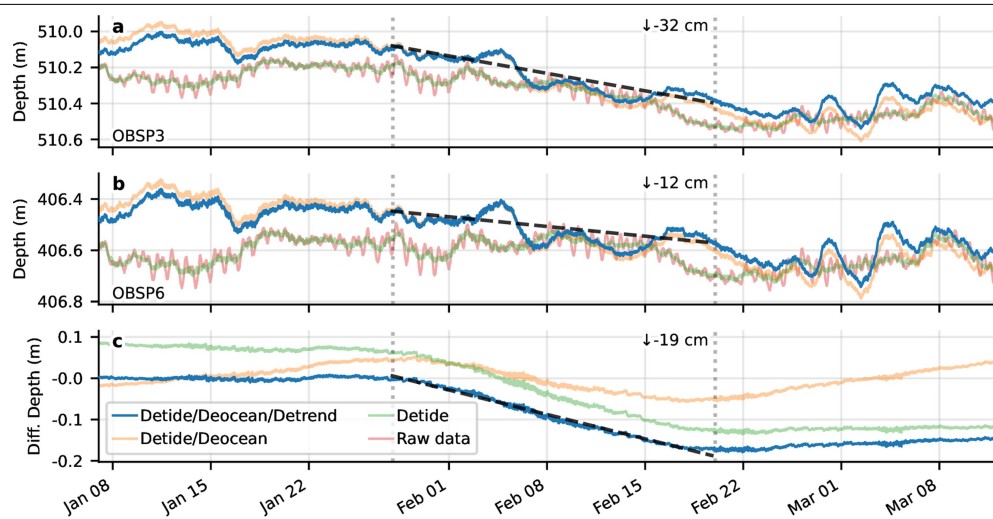

**Extended Data Fig. 6 | Surface displacement from ocean-bottom pressure sensors. a**, Pressure recordings converted to depth and processed time series of OBSP3 recorded within the crater of Kolumbo showing a subsidence of -32 cm. **b**, Pressure recordings converted to depth and processed time series of OBSP6 recorded on the northern flank of Kolumbo showing a subsidence of -12 cm. **c**, Relative depth changes of -19 cm between OBSP3 and OBSP6.

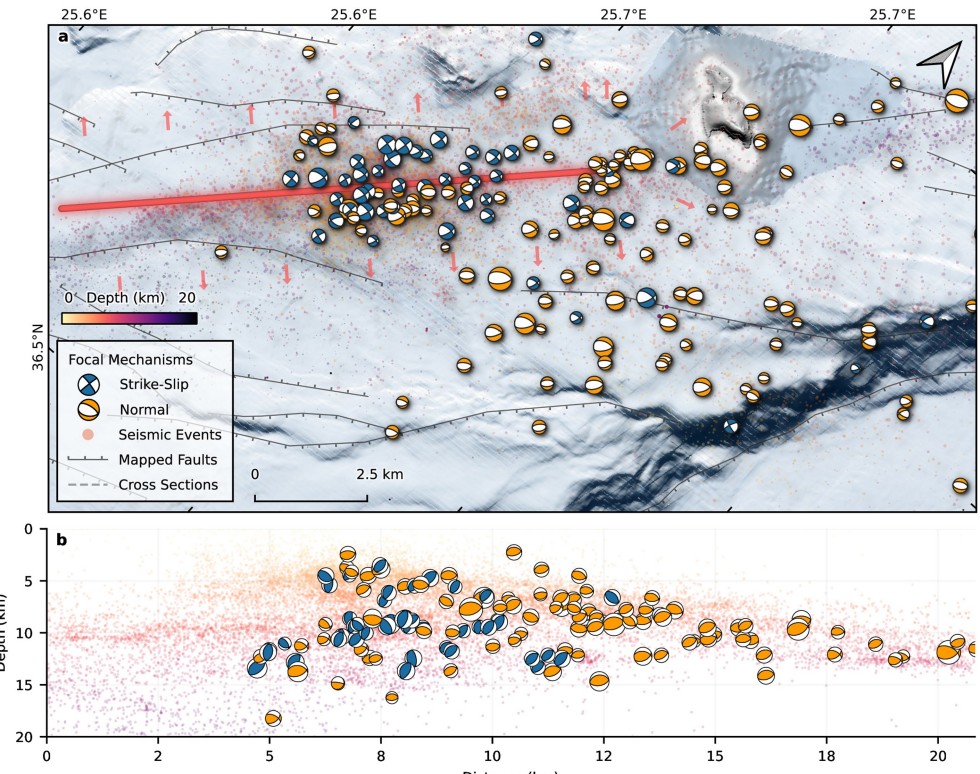

**Extended Data Fig. 7 | Spatial distribution of focal mechanisms. a**, Close-up map focusing on the earthquake locations and focal mechanisms of strike-slip (blue) and normal faulting events (orange) SE of Anhydros. The modelled dike intrusion is depicted in red. **b**, Depth cross-section of the map view with in-view projected focal mechanisms.

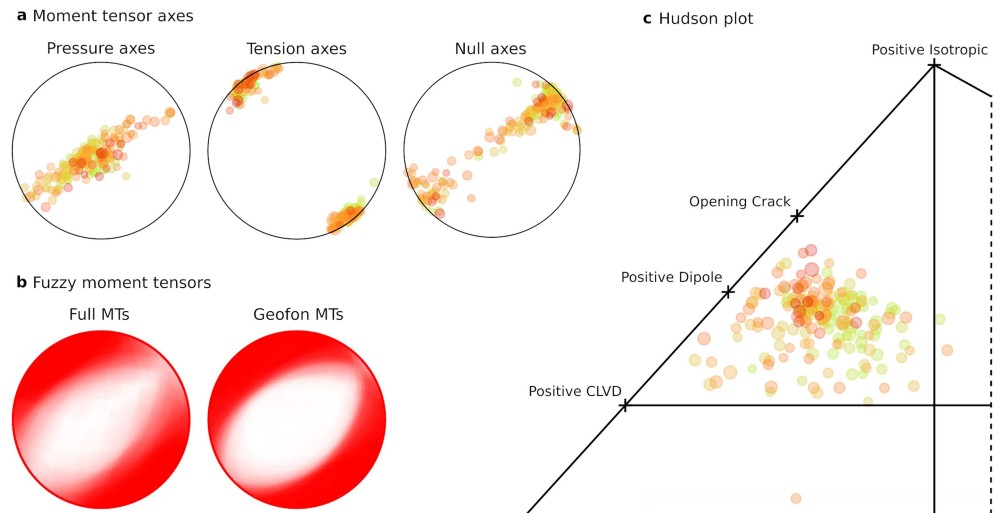

**Extended Data Fig. 8 | Source mechanisms during the swarm/diking. a**, Polar distribution of pressure, tension and null axes, **b**, Comparison of fuzzy moment tensors (MTs), overlap of focal spheres for each event (best full MTs and Geofon routine MTs). **c**, The Hudson plot shows consistent positive isotropic and compensated linear vector dipole (CLVD) components, denoting a combination of shear and opening tensile failure. Each solution is represented by a circle, with size proportional to the magnitude and colour denoting time (colour scale in Fig. 1). Comparison of moment tensor solutions from operational monitoring at GEOFON and this study's full waveform inversion.

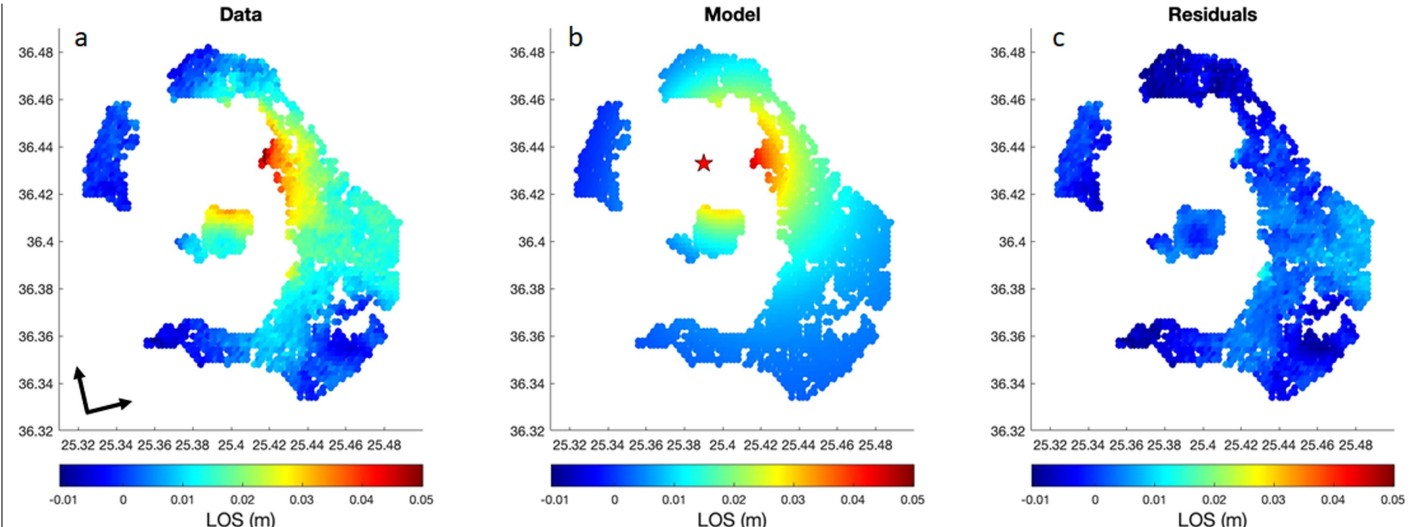

**Extended Data Fig. 9 | Geodetic model for intra-caldera inflation.** The model covers the time period 10 July 2024 to 18 January 2025. **a**, Line-of-sight (LOS) displacements from Sentinel−1 ascending track 029. **b**, Best-fit model prediction using an inflating point source (Mogi, 1958) (red star). **c**, Residuals.

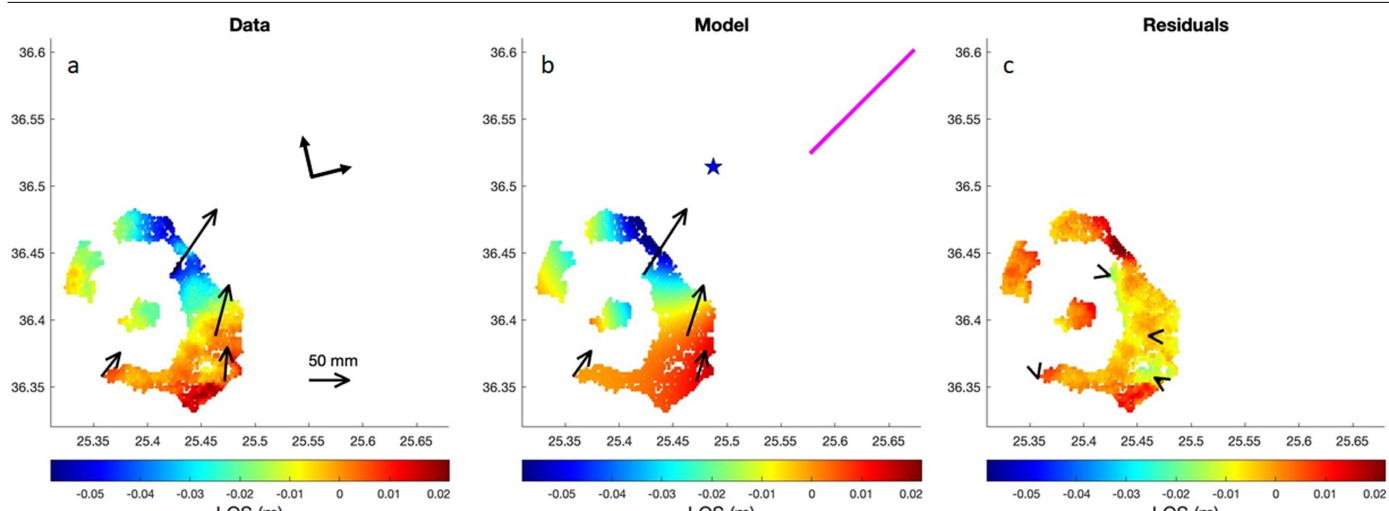

**Extended Data Fig. 10 | Geodetic model for the diking period.** The model covers the time period 18 January to 23 February 2025. **a**, Line-of-sight (LOS) displacements from Sentinel-1 ascending track 029 and horizontal GNSS displacements (black arrows). **b**, Best-fit model prediction using a deflating point source (Mogi, 1958) (blue star) and inflating dike (Okada, 1992) (magenta line). **c**, Residuals.

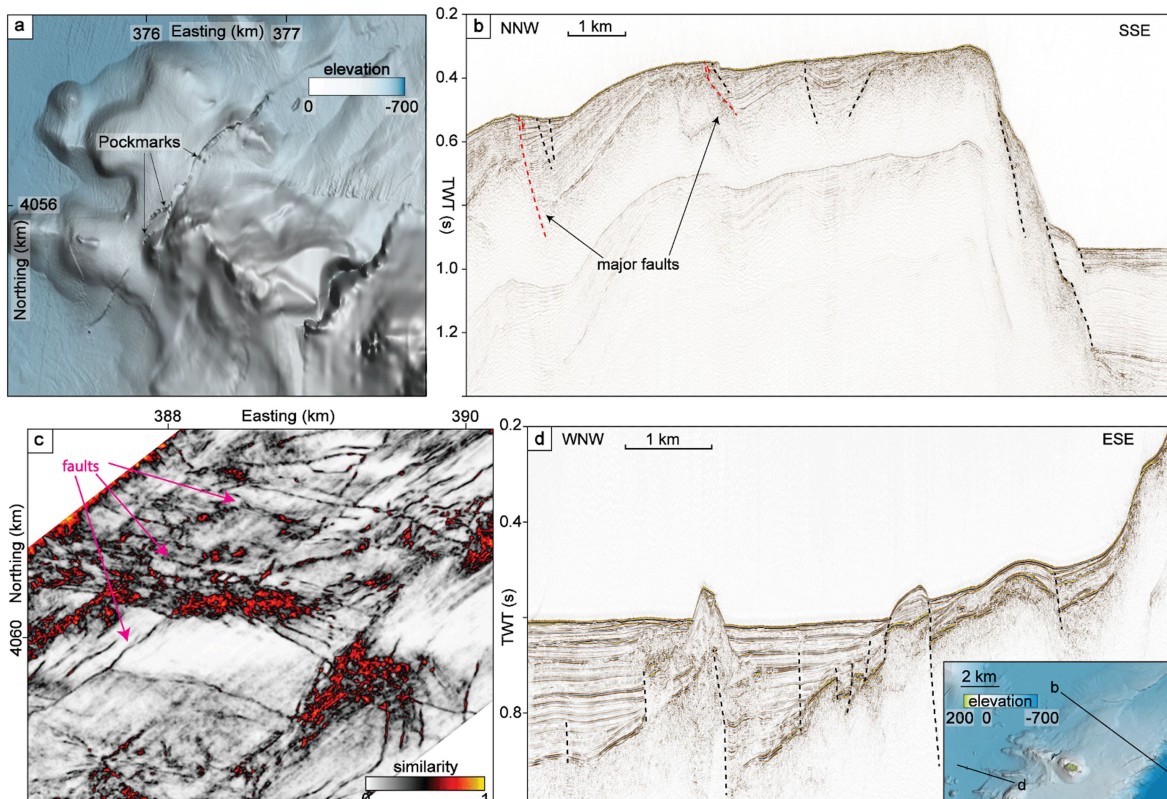

**Extended Data Fig. 11 | Bathymetric and seismic data. a**, High-resolution bathymetry data of Amorgos faults showing pockmarks. The coordinates are in UTM Zone 35 N. **b**, High-resolution seismic profile crossing the relay ramp structure of the Amorgos Fault. The orange line shows the two-way time of the surface shown in panel **c. c**, Similarity time slice at 550 ms. **d**, High-resolution seismic profile cutting the Anhydros Basin and a cone of the KVC.