## [Peer Review File · Nature]

Volcanic crisis reveals coupled magma system at Santorini and Kolombo

Corresponding Author: Dr Marius Isken

Version 1:

Reviewer comments:

Referee #1

(Remarks to the Author)

Thank you for the opportunity to review the manuscript titled “Volcanic crisis reveals coupled magma system at Santorini and Kolombo” by Isken et al. This study documents a recent and remarkable episode of geophysical unrest in the Kolombo Volcanic Chain (KVC) - initially interpreted as a tectonic sequence - and presents compelling evidence that this unrest was instead driven by magma intrusion within a plumbing system linking Santorini and Kolombo. The topic is novel and significant, and the data analysis is of high quality. I believe the study has the potential to be published in Nature.

However, I encourage the authors to revise the manuscript with greater attention to the relationship between their seismic and geodetic datasets. This crisis presents a relatively rare opportunity to study how magma intrusion generates volcano-tectonic swarms, and the high-resolution earthquake catalog deserves a more integrated discussion alongside the modeled dike intrusions. There are also numerous typographical and presentational issues, detailed below as minor comments.

Major Comments

- Lines 56–57: “characteristic pattern of hypocentre migration”

Hypocenter migration is not a globally characteristic feature of volcano-seismic swarms. While well-known cases exist (e.g., La Palma, Kīlauea, Holuhraun), many (most?) dike-related swarms - including those preceding eruption - show no clear migration patterns, even when relocated precisely. (For a discussion, see Roman & Cashman, 2018.) This is not a citation request(!), but a caution against reinforcing a common misconception. Cases like Mt. St. Helens (2004) and Okmok (2008) demonstrate that swarms can occur without spatial migration (e.g., Moran et al., 2008; Garza-Giron et al., 2023). Where hypocenter migration does occur, it is generally associated with low-viscosity magma and/or an extensional regime- both of which appear relevant here.

A simple revision would be to remove the word “characteristic” from this sentence. However, if space permits, the manuscript would benefit from a deeper exploration of why the 2025 KVC swarm does show migration (e.g., extensional tectonics, magma properties).

- Lines 124–127: Regarding focal mechanisms and extensional fracturing

The statement that the observed mechanisms reflect “rupture patterns associated with extension fractures common during dike intrusions...” overgeneralizes. Globally, strike-slip and even thrust faulting appear to be more common during dike-related swarms than extensional faulting, outside of specific regions like Iceland or Kīlauea.

- Figure 2c does not show strike-slip events, despite their presence in the PT axes in Figure 1b and in the quoted text. If this is due to visual overlap, the figure should be clarified.

- It is essential to show the spatial distribution of the 180 moment tensor (MT) events. Where did the strike-slip and normal-faulting events occur relative to the modeled dike intrusions (e.g., Fig. 3c)?

- The interpretation that “normal faulting occurs above and below propagating dikes, and strike-slip faulting emerges at their tips and tails” is interesting but speculative without corresponding spatial evidence. This makes it even more important to show the MT event locations in context with the geodetic models.

Minor Comments

- Line 211: Also Nyiragongo/Nyamulagira.
- Line 221: Also Ambrym (see Shreve et al. 2019).
- Figure 1: Add GPS stations (dots with vectors) to the legend in panel (a).
- Figure 2: Rotate the depth scale bar in panel (a) to vertical. In panel (c), clarify what the MT symbol sizes represent (presumably magnitude). Indicate this in the caption.
- Figure 3: Label panel (c) with the time periods of each modeled dike. While this information appears in a supplementary figure, showing it here would help readers interpret the spatial and temporal evolution.
- Figure 4b: There is a mismatch between the dike depths shown here and those in Figure 3c. Figure 4b implies a single dike at 8–14 km depth, while Figure 3c shows shallower, eastward-progressing intrusions reaching up to 4 km. Consider adjusting the figure or clarifying the perspective to reconcile these views.
- Supplementary Figures:
 - There are two figures labeled “Supplementary Figure 1,” and the final set of figures (seismicity migration steps) lacks a caption.
 - The first Supplementary Figure includes the only map showing MT locations, but the scale is too broad to discern detail. See major comments above.
 - The second Supplementary Figure repeats the date range 2025-02-04 to 2025-02-12.
 - In the third Supplementary Figure, Santorini is hard to locate—please label it or zoom out for context.
- Extended Data Figure Numbering: Please reorder to match the order of first appearance in the manuscript. For example, EDF 7 is referenced before EDFs 1 and 2.
- Extended Data Figure 6: The caption is confusing. It reads:
“a, The full waveform inversion using ground and, b, focal mechanisms calculated by GEOFON. The moment tensor’s pressure and tension axis are comparable and indicate the strike and opening of the dike intrusion.”
Clarify what “a” and “b” refer to (currently both appear to relate to moment tensors). If needed, relabel the panels for clarity.
- Extended Data Video 1: Please add a legend (color scale, etc.) and label key geographic features. A north arrow would help. Also: are MT events included? If not, consider adding them, or clarify how they relate to the features shown.

Kind regards,
Diana Roman

Referee #2

(Remarks to the Author)
see attached

Version 2:

Reviewer comments:

Referee #1

(Remarks to the Author)

Thank you for the opportunity to review the revised version of the manuscript “Volcanic crisis reveals coupled magma system at Santorini and Kolombo” by Isken et al. The authors have done an excellent job in addressing all of my concerns regarding the original version of this manuscript, and I can now recommend this manuscript for publication in Nature pending a few minor typographical errors. This is a compelling and important study, and I commend the authors on their efforts.

Minor revisions:

Figure 4 - L3 label is shown twice (this is correct as shown in Autumn et al 2025), but may be confusing to readers (I initially thought one of the 'L3' labels was a mistake and should have been 'L1' or 'L2'). Perhaps a short note in the caption could be added to clarify.

Figure 3 caption - "Vertical dashed lines mark the temporal phases of seismicity (Fig. 2)." has been added to the 3c caption but I think it actually refers to panel d(?) Move this sentence to end of caption.

"Mikajima" -> "Miyakejima" (at the end of the second-to-last paragraph in section "Spatiotemporal evolution")

Kind regards,
Diana Roman

Referee #2

(Remarks to the Author)

Isken et al, revised: Volcanic crisis reveals coupled magma system at Santorini and Kolumbo

The revised manuscript addresses most of the issues raised in the initial reviews. In particular, the interpretation of the source region for the dikes is now made clearer. Also, the uncertainty in the depth of the deflating source, and its connection to the rest of the magmatic system has been clarified. While the paper is acceptable at this point, there are two points that I

think could be improved.

In my initial review I noted “ Why should there be a threshold injection rate for eruption? The next sentence says the system hasn’t reached a threshold volume change. Physically, neither makes much sense – chamber overpressure relative to the local least compressive stress makes the most physical sense.” The authors refer to a paper by Browning et al (2015) to support the idea of an injection threshold and the magnitude of this threshold. I looked at this paper provides a “back of the envelope” calculation that makes rather strong, and possibly unrealistic, assumptions. I don’t think the authors are doing themselves a service to pin any interpretations on this estimate and would suggest omitting it entirely.

The second point has to do with appealing to local and regional poroelastic stress changes to explain seismicity within the faulted basement. It is not clear what the authors mean here by “poroelastic”. Do you mean undrained poroelastic response, where there are stress changes induced by the dike opening, without any flow of pore-fluids in the basement rocks? Or are you envisioning that magmatic fluids are permeating out of the dike into the adjacent crust thereby altering the effective mean normal stress state? In the general theory of poroelasticity, changes in pore pressure induced deformations, and changes in deformation induce pore pressures. The usage in the manuscript is unclear.

As I said the paper is basically acceptable, so I leave it to the editors and authors to work out any further changes.
Paul Segall

Response Letter

Potsdam, 2. July 2025

Nature - Manuscript 2025-02-04618A

Dear Editor, Dear Reviewers,

Thank you for considering our manuscript “*Volcanic crisis reveals coupled magma system at Santorini and Kolumbo*” for publication in Nature. We would like to thank Diana Roman and Paul Segall for their time and valuable and constructive review, which significantly improved the quality and accessibility of our work.

The revised manuscript addresses the insights and concerns raised by the reviewers, providing a more precise presentation and discussion of our research.

Following the recommendations of the reviewers, we made the following main changes to the manuscript:

1. Improved discussion of the seismicity and focal mechanisms during migration of the dike. Added Extended Data Fig. 7 showing map and cross-section focusing on the moment tensor evolution and clustering. Emphasising the link between independent geodetic and seismic observations.
2. Improved integrated discussion of the magma migration paths and poroelastic effects. Improvement of conceptual interpretation, Figure 4.

Minor changes include:

1. Improved readability throughout the manuscript (minor wording changes), without altering the meaning.
2. Improved figures in the main text for reader accessibility and clarity.

Below, you will find a detailed point-by-point response.

Please find below all comments by the editor and reviewers, followed by our reply in **semibold text**.

With warm regards,

Marius Isken and Jens Karstens

Reply to Reviewers

Referee #1

Thank you for the opportunity to review the manuscript titled “Volcanic crisis reveals coupled magma system at Santorini and Kolombo” by Isken et al. This study documents a recent and remarkable episode of geophysical unrest in the Kolombo Volcanic Chain (KVC) - initially interpreted as a tectonic sequence - and presents compelling evidence that this unrest was instead driven by magma intrusion within a plumbing system linking Santorini and Kolombo. The topic is novel and significant, and the data analysis is of high quality. I believe the study has the potential to be published in Nature.

Thank you for the insightful and constructive review of our manuscript.

However, I encourage the authors to revise the manuscript with greater attention to the relationship between their seismic and geodetic datasets. This crisis presents a relatively rare opportunity to study how magma intrusion generates volcano-tectonic swarms, and the high-resolution earthquake catalog deserves a more integrated discussion alongside the modeled dike intrusions. There are also numerous typographical and presentational issues, detailed below as minor comments.

We agree with this comment. We emphasise the agreement between the two independent yet complementary results from geodetic modelling and seismic observations. The improved discussion is now cross-referencing geodetic modelling and seismic observations.

We thank the reviewer for this valuable suggestion and fully agree that the relationship between the seismic and geodetic datasets deserves a more integrated and focused discussion. We have revised the manuscript accordingly to more clearly highlight the interplay between these two independent yet complementary datasets.

In particular, we have significantly improved the text in the Discussion (now labelled “Dynamics, interactions and coupling”; page 10, paragraph 3), where we now explicitly correlate the phases of dike intrusion inferred from geodetic modelling with the spatiotemporal evolution of the high-resolution earthquake catalogue. This includes a detailed comparison of modelled dike paths and depths with the observed seismicity clusters and anomalies in seismic velocity structure, particularly in relation to anomalies L3, L4, and H1 (described by Hufstetler et al., 2025 and Autumn et al., 2025). The revised paragraph now reads as follows:

“The 2025 Santorini-Kolumbo dike intrusion nucleated in the direct vicinity of a low P-wave velocity (V_p) anomaly L4 (Fig. 4) previously identified through seismic tomographies^{31,32}. Low- V_p anomalies L3 and L4 extend from east of Santorini and south Kolumbo upward to the NE beneath the Anhydros block and represent a branching lower- to mid-crustal magma storage system containing at least 5-14% partial melt^{31,32}. Based on our high-resolution seismicity catalogue, the dike intrusion initiated at approximately 18 km depth, while deeper magma migration was likely aseismic⁴¹. First, the dike ascended through the low- V_p anomaly L4 (Phase I). Then it migrated shallower and to the northeast, wrapping around storage

zone L3 and focusing above L3 at ~5 km depth (Phases II-V), which marks the phase of highest inflow rates (Fig. 3c). The subhorizontal dike migration during Phase IV lies within the deepest and northeastern-most extent of L3. The modelled location of co-diking magma depletion at 7.5 ± 1 km depth beneath Kolumbo correlates with both (1) a concurrently occurring cluster of vertically oriented seismicity at 8.5-13 km depth and (2) the base of the high-Vp/low-Vs (S-wave velocity) anomaly H1, previously interpreted as a rheologically strong layer with vertical, melt-filled cracks^{15,31}. These combined observations suggest that magma is transferred vertically between separate branches of the mid-crustal storage system, implying a vertical hydraulic linkage between the deflation source and the dike-feeding system.”

Further, we added the following sentence to the discussion:

“The 2025 Santorini crisis offers an unprecedented opportunity to study the activated volcanic system through independent observations of geodetic deformation and seismicity. The sequential timing of shallow magma [...]”

The final paragraph of the manuscript highlights the necessity of high-resolution catalogues for understanding volcanic systems.

Figure 3c now displays the seismic phases as an important cross-reference between spatiotemporal seismicity and geodetic modelling.

*Lines 56–57: “characteristic pattern of hypocentre migration”
Hypocenter migration is not a globally characteristic feature of volcano-seismic swarms. While well-known cases exist (e.g., La Palma, Kilauea, Holuhraun), many (most?) dike-related swarms - including those preceding eruption - show no clear migration patterns, even when relocated precisely. (For a discussion, see Roman & Cashman, 2018.) This is not a citation request(!), but a caution against reinforcing a common misconception. Cases like Mt. St. Helens (2004) and Okmok (2008) demonstrate that swarms can occur without spatial migration (e.g., Moran et al., 2008; Garza-Giron et al., 2023). Where hypocenter migration does occur, it is generally associated with low-viscosity magma and/or an extensional regime - both of which appear relevant here. A simple revision would be to remove the word “characteristic” from this sentence. However, if space permits, the manuscript would benefit from a deeper exploration of why the 2025 KVC swarm does show migration (e.g., extensional tectonics, magma properties).*

We agree that this statement is overly generalised. Therefore, we have removed the word “characteristic” when describing observations.

We have improved the description of seismicity migration and added references to relevant studies about the characteristics and migration of seismicity during diking (see quotation on page 7 of the response letter).

Lines 124–127: Regarding focal mechanisms and extensional fracturing. The statement that the observed mechanisms reflect “rupture patterns associated with extension fractures common during dike intrusions...” overgeneralizes. Globally, strike-slip and even thrust faulting appear to be more common during dike-related swarms than extensional faulting, outside of specific regions like Iceland or Kīlauea.

We agree that this statement was also too generalised. Please see the following comment for details of how we have improved the discussion and figures.

Figure 2c does not show strike-slip events, despite their presence in the PT axes in Figure 1b and in the quoted text. If this is due to visual overlap, the figure should be clarified. It is essential to show the spatial distribution of the 180 moment tensor (MT) events. Where did the strike-slip and normal-faulting events occur relative to the modeled dike intrusions (e.g., Fig. 3c)? The interpretation that “normal faulting occurs above and below propagating dikes, and strike-slip faulting emerges at their tips and tails” is interesting but speculative without corresponding spatial evidence. This makes it even more important to show the MT event locations in context with the geodetic models.

We agree. We have improved the discussion, added references to dike-related migrations of seismicity, improved existing figures, and added Extended Data Fig. 7.

“The observed focal mechanisms range from normal faulting to strike-slip and oblique faulting, typical in extensional settings. Most events exhibit non-double-couple terms with positive isotropic and compensated linear vector components, indicating complex, dilational processes accompanying shear faulting. Phases of lateral migration of seismicity (II and IV) are dominated by normal faulting, episodes of upward propagation of seismicity (III and the beginning IV) are characterised by a notable increase of low-magnitude strike-slip events that focus on an area SE of Anhydros (Fig. 3c, Extended Data Fig. 7). Such spatiotemporal evolution resembles other dike intrusions in extensional settings, including Mikajima⁵, Iceland²⁶, and Kīlauea⁶.”

Fig. 2c has been updated to highlight the temporal evolution of double-couple focal mechanisms. This improves the discussion of the dike migration pattern. Furthermore, we have added a map and cross-section displaying the spatial distribution of focal mechanisms with a focus on the dike centre to the southeast of Anhydros (see Extended Data Fig. 6). Both figures more clearly illustrate the normal faulting associated with phases of dike progression and strike-slip faulting during dike inflation.

Line 211: Also Nyiragongo/Nyamulagira. Line 221: Also Ambrym (see Shreve et al. 2019).

Nyiragongo and Ambrym are additional relevant examples, but we have decided not to include them as we have already reached the maximum number of references. We now reference Wright et al. (2012; <https://doi.org/10.1038/ngeo1428>): A comprehensive study of dike propagation.

Figure 1: Add GPS stations (dots with vectors) to the legend in panel (a).

Done. The legend has been updated to include GNSS stations and arrows, as well as the legend in Figure 3.

Figure 2: Rotate the depth scale bar in panel (a) to vertical. In panel (c), clarify what the MT symbol sizes represent (presumably magnitude). Indicate this in the caption.

Done. The depth colorbar is vertical. The moment tensors have been reduced to DC and coloured according to their mechanism (strike-slip or normal faulting). The updated caption reflects the sizes of the moment-tensor symbols.

Figure 3: Label panel (c) with the time periods of each modeled dike. While this information appears in a supplementary figure, showing it here would help readers interpret the spatial and temporal evolution.

Done. We added numbers to the modelled Okada planes in Fig. 3c to better illustrate the succession and migration of the modelled dike.

Figure 4b: There is a mismatch between the dike depths shown here and those in Figure 3c. Figure 4b implies a single dike at 8–14 km depth, while Figure 3c shows shallower, eastward-progressing intrusions reaching up to 4 km. Consider adjusting the figure or clarifying the perspective to reconcile these views.

We agree. The dike plane obstructing the seismicity cloud has been removed from conceptual Figure 4b. Additionally, we labelled the structural elements in accordance with Hufstedtler et al. (2025) and Autumn et al. (2025).

Supplementary Figures: There are two figures labeled “Supplementary Figure 1,” and the final set of figures (seismicity migration steps) lacks a caption.

Done. The numbering of the supplementary figures has been corrected.

◦ *The first Supplementary Figure includes the only map showing MT locations, but the scale is too broad to discern detail. See major comments above.*

Done. Figure 2c has been updated, and Extended Data Figure 7, which focuses on the spatiotemporal evolution of focal mechanisms associated with dike emplacement, has been added. Supplementary Figure 1 illustrates the network geometry used for the moment tensor inversion.

◦ *The second Supplementary Figure repeats the date range 2025-02-04 to 2025-02-12.*
◦ *In the third Supplementary Figure, Santorini is hard to locate—please label it or zoom out for context.*

Done. We updated and corrected the figures. We have created new supplementary figures showing the migration of seismicity in seven distinct time windows.

Extended Data Figure Numbering: Please reorder to match the order of first appearance in the manuscript. For example, EDF 7 is referenced before EDFs 1 and 2.

Done. We rearranged the Extended Data Figures.

Extended Data Figure 6: The caption is confusing. It reads: "a, The full waveform inversion using grond and, b, focal mechanisms calculated by GEOFON. The moment tensor's pressure and tension axis are comparable and indicate the strike and opening of the dike intrusion." Clarify what "a" and "b" refer to (currently both appear to relate to moment tensors). If needed, relabel the panels for clarity.

Corrected. We have removed the stray sentences from the caption.

Extended Data Video 1: Please add a legend (color scale, etc.) and label key geographic features. A north arrow would help. Also: are MT events included? If not, consider adding them, or clarify how they relate to the features shown.

We improved the animation and added a colour bar, as well as a Cartesian grid to improve 3D perception. We added geographical labels and highlighted the islands' outlines.

Updated animation: <https://youtu.be/E4ONsNy7xoE>

Referee #2

This paper reports on a fascinating seismic sequence in the vicinity of Santorini, an active volcanic island in the Mediterranean, which will be of significant interest to Nature readers. The paper reports on new seismic, geodetic, and other observations which are relevant to understanding the process that gave rise to the sequence. In general, the paper is well written and suitable for publication. I do have a few suggestions which I believe will improve the manuscript.

Thank you for your insightful and constructive comments, which we address in the manuscript and below.

The interpretation put forward by the authors is that there were a series of dikes propagating largely to the northeast, starting somewhere below Santorini. This is supported by the earthquake locations and that limited GNSS data. The dike injections were preceded by inflation at Santorini and accompanied by subsidence measured by a sea-floor pressure gauge at Kolumbo volcano. This latter is interpreted as resulting from outflow from a shallow reservoir. It is not stated how well constrained the depth of this reservoir is, nor what features in the data control this part of the model. Is this feature controlled by the single pressure gauge? What controls its depth?

Multiple seismic experiments have been conducted to constrain the reservoirs beneath Kolumbo. Full-waveform inversion identified a shallow magma chamber at a depth of ~2 km beneath Kolumbo (Chrapkiewicz et al., 2022). Meanwhile, tomographic experiments indicate the presence of a mid-crustal magma reservoir reaching depths of up to 8 km, identified from a low P-wave velocity anomaly known as L3 (6 - 8 km) and L4 (10 - 20 km; Hufstetler et al., 2025, Autumn et al., 2025).

Our initial deformation models were inverted without the OBSP data, as the data only became available in mid-March following their recovery. Earlier simulations without the OBSPs resulted in Mogi deflation source locations approximately 1.5 km deeper and around 2 km south of the locations obtained from inversions that included the OBSPs.

We are aware of the uncertainties in our inversions and provide details about the model's confidence intervals in the caption of Figure 3. To provide readers with more detailed information, we have added posterior probability density plots to the Supplementary Material, illustrating the uncertainty of each inverted parameter for the pre-diking and diking phases.

We broadened the discussion, and Figure 4a has been updated accordingly:

“The 2025 Santorini-Kolumbo dike intrusion nucleated in the direct vicinity of a low P-wave velocity (V_p) anomaly L4 (Fig. 4) previously identified through seismic tomographies^{31,32}. Low- V_p anomalies L3 and L4 extend from east of Santorini and south Kolumbo upward to the NE beneath the Anhydros block and represent a branching lower- to mid-crustal magma storage system containing at least 5-14% partial melt^{31,32}. Based on our high-resolution seismicity catalogue, the dike intrusion initiated at approximately 18 km depth, while deeper magma migration was likely aseismic⁴¹. First, the dike ascended through the low- V_p anomaly L4 (Phase I). Then it migrated shallower and to the northeast, wrapping around storage zone L3 and focusing above L3 at ~5 km depth (Phases II-V), which marks the phase

of highest inflow rates (Fig. 3c). The subhorizontal dike migration during Phase IV lies within the deepest and northeastern-most extent of L3. The modelled location of co-diking magma depletion at 7.5 ± 1 km depth beneath Kolumbo correlates with both (1) a concurrently occurring cluster of vertically oriented seismicity at 8.5-13 km depth and (2) the base of the high-Vp/low-Vs (S-wave velocity) anomaly H1, previously interpreted as a rheologically strong layer with vertical, melt-filled cracks^{15,31}. These combined observations suggest that magma is transferred vertically between separate branches of the mid-crustal storage system, implying a vertical hydraulic linkage between the deflation source and the dike-feeding system.”

The discussion section notes that there is deep seismicity below Kolumbo; however, the connection between the subsidence and the deep seismicity is not clear and the depth separation is quite large.

The earlier version of Figure 4 was misleading because the deflation source was illustrated as being shallower than modelled. However, the inverted depth of the deflation (~ 7.5 km ± 0.7 km) and the depth of the seismicity clusters, which we propose are connected to magma depletion at 8.5–13 km, fit together remarkably well. This is also consistent with the tomography results. This is particularly noteworthy given that Mogi sources usually underestimate the depth of deflations. We have updated Figure 4 accordingly.

Moreover, the origin of the magma for the dikes is unresolved. It does not appear to be from a shallow Santorini magma reservoir, and the paper cites petrologic evidence for different source magmas at Santorini and Kolumbo. The schematic in Figure 4 shows an intermediate depth magma accumulation zone. The seismicity, if anything, seems to wrap around and above this zone. Is this the source of the magma?

We improved the integrated discussion (see above answer), clarifying the origin and migration of the dike based on a tomographic regional framework.

The dike most likely originates from the deep reservoir (see reply above). The source zone of the dike is located 18 km between Santorini and Kolumbo and has not been imaged by any seismic experiment. However, it is unlikely that the dike's source is the comparatively small and shallow reservoirs of Santorini and Kolumbo, which are located at depths of ~ 3 km. Autumn et al. (2025) provided a detailed discussion of the potential for deeper migration pathways and feeding structures for both volcanoes, which could be either linked or entirely separate. Our results show coupling between the two systems at mid-crustal depths between the two centres (Response Fig. 1). However, since we cannot image deep aseismic magma migration, which likely plays an important role in the deeper crust (as illustrated, for example, by del Fresno et al., 2023, for La Palma), we cannot resolve the deep feeding system in enough detail to elaborate further on the proposed hypothesis. Nevertheless, our updated discussion provides a more specific explanation of the feeding system by integrating the results of Autumn et al. (2025) and Hufstetler et al. (2025) more thoroughly.

[REDACTED]

Response Fig. 1: The conceptual model hypothesis of the drainage system beneath Santorini/Kolumbo, from Hufstedler et al., 2025. The findings of our study support Hypothesis 1, emphasising the connection between a mid-crustal magma chamber and the Santorini and Kolumbo volcanoes.

Elsewhere, in a quite speculative statement, much of the seismicity is ascribed to dike-induced pore-pressure changes in the surrounding crust. This and other statements suggest a significant role for tectonic faulting. While I understand that the authors do not have all the RESPONSEs, I think it would be appropriate to more clearly indicate which aspects of the interpretation are strongest and where there is considerable uncertainty, for example in the relative role of tectonic seismicity.

We agree that induced poroelastic stresses are the dominant factor, rather than fluid flows and pore pressure alone. We have corrected and improved the wording to explicitly indicate dike-induced seismicity and poroelastic stress changes that trigger earthquake activity on pre-existing structures within this extensional regime. We rephrased the text and added information about the uncertainty of seismicity at the end of paragraph 1 on page 11.

“The dike intrusion caused local and regional poroelastic stress changes, inducing and triggering seismicity⁴² within the highly faulted basement of the Anhydros block. The extent to which the normal-faulting earthquakes have released tectonic pre-stress due to the dike’s emplacement remains unknown.”

Smaller points that will clarify the manuscript. Page 2, paragraph 2, refers to subsidence at Kolumbo crater but doesn’t explain where that is located. Should refer to Figure 3 where it is identified.

We added '*located*' to the sentence to clarify the location of the swarm initialisation.

Figure 3c is supposed to show the temporal evolution of the dikes, here represented as rectangular dislocations. However, there is nothing to indicate the temporal progression. Could this be represented by color in some way, as is done in the supplement?

We agree. We have updated Fig. 3c and added numbers to the Okada sources modelled consecutively to indicate their temporal progression.

The time-dependent dike model is, as I understand, a sequence of time intervals with best-fitting rectangular dislocations. There is thus no physical connection between time steps. The smoothing is visually appealing but was not used in the modeling to predict deformation data.

We agree and have reduced the smoothing kernel to 0.2 km to represent the rectangular sources. We added time-step numbers to the Okada dislocation models. We switched to a colour map to enhance the representation of zero openings.

Discussion, first paragraph: Why should there be a threshold injection rate for eruption? The next sentence says the system hasn't reached a threshold volume change. Physically, neither makes much sense – chamber overpressure relative to the local least compressive stress makes the most physical sense.

We were imprecise regarding the threshold. Browning et al. (2015) investigated past dike intrusions at Santorini by measuring dike dimensions in the field. From these measurements, they inferred the volume of magma inflow required to trigger an eruption. We changed the sentence to “inflow volume threshold” (page 11, paragraph 1) to be more specific. The study by Browning et al. (2015) is the most frequently cited analysis of the boundary conditions controlling diking at Santorini. While we agree that local compressive stress generally plays a major role, we believe it is important to mention the rupture criteria suggested by Browning et al. (2015) in the context of the 2025 events (see Response Fig. 2). However, a recent study by Galetto et al. (2022) statistically investigated the correlation between magma injection rates and eruptions in caldera systems. The key outcome of this study was that the critical injection rate for an eruption is proportional to the size of the magma chamber. However, this study examined various calderas on Earth and did not focus on Santorini; therefore, we do not cite it in the revised manuscript.

[REDACTED]

Response Fig. 2: Excess pressure (p_e) within the shallow magma chamber at Santorini as a function of the volume of new magma (ΔV_m) entering the chamber from a deeper source

over time. Modified from Browning et al., 2015.

Page 10, top. The sentence “The dike-induced local stress changes induced fluid flow and pore pressure changes within the highly faulted basement of the Anhydros block and triggered secondary tectonic earthquakes, illuminating broad structural elements” is quite speculative. It may be plausible but, unless there is direct evidence for this, it should be identified as speculation.

Yes, we agree. We have refined the discussion and are now focusing on primary induced and secondary triggered events through poroelastic stress changes. Please see the response above for more information.

“The dike intrusion caused local and regional poroelastic stress changes, triggering seismicity within the highly faulted basement of the Anhydros block. The extent to which the normal-faulting earthquakes have released tectonic pre-stress due to the dike’s emplacement remains unknown.”

Figure 4a needs reference for interpretation as mid-crustal magma reservoir.

Yes, we agree. Figure 4a has been updated to label the features of the underlying tomography from Hufstetler et al. (2025). See previous response.

Relevant References

del Fresno, C., Cesca, S., Klügel, A. et al. Magmatic plumbing and dynamic evolution of the 2021 La Palma eruption. *Nat Commun* 14, 358 (2023). <https://doi.org/10.1038/s41467-023-35953-y>

Galetto, F., Acocella, V., Hooper, A. et al. Eruption at basaltic calderas forecast by magma flow rate. *Nat. Geosci.* 15, 580–584 (2022). <https://doi.org/10.1038/s41561-022-00960-z>

Browning, J., Drymoni, K. & Gudmundsson, A. Forecasting magma-chamber rupture at Santorini volcano, Greece. *Sci Rep* 5, 15785 (2015).

Segall, P., and S. Lu (2015), Injection-induced seismicity: Poroelastic and earthquake nucleation effects, *J. Geophys. Res. Solid Earth*, 120, 5082–5103, doi:10.1002/2015JB012060.

Wright, T., Sigmundsson, F., Pagli, C. et al. Geophysical constraints on the dynamics of spreading centres from rifting episodes on land. *Nature Geosci* 5, 242–250 (2012). <https://doi.org/10.1038/ngeo1428>

Response Letter

Potsdam, 11. August 2025

Nature - Manuscript 2025-02-04618C

Dear Editor, Dear Referees,

Thank you for accepting our manuscript “Volcanic crisis reveals coupled magma system at Santorini and Kolumbo” for publication in Nature. We find the referee’s comments valuable and address them in the attached revised manuscript.

Minor changes include:

1. Corrected typography.
2. Improved discussion based on referee’s comments:
 - a. Removed notion of eruption threshold.
 - b. More concrete discussion of poroelastic stress changes during dike emplacement.
3. We added (swapped) a new reference to strengthen the argument for the vertical hydraulic communication of magma between the systems (ref 42).
4. Improved *Extended Data Figure 6*.
5. Added *Supplementary Data 1* (Earthquake catalog).

Below, you will find a detailed point-by-point response with our reply in **bold text**.

This is a co-lead authorship between Marius Isken and Jens Karstens. We wish to be cited as:

Isken, M. P. and Karstens, J. *et al.* Volcanic crisis reveals coupled magma system at Santorini and Kolumbo ...

We wish to participate in a transparent peer review process.

With best regards,

Marius Isken and Jens Karstens

Reply to Reviewers

Referee #1 (Diana Roman)

Thank you for the opportunity to review the revised version of the manuscript “Volcanic crisis reveals coupled magma system at Santorini and Kolombo” by Isken et al. The authors have done an excellent job in addressing all of my concerns regarding the original version of this manuscript, and I can now recommend this manuscript for publication in Nature pending a few minor typographical errors. This is a compelling and important study, and I commend the authors on their efforts.

Thank you again for the thorough review of the manuscript; your input from this and the previous round of reviews significantly improved the clarity and helped shape the scientific perspective.

Minor revisions:

Figure 4 - L3 label is shown twice (this is correct as shown in Autumn et al 2025), but may be confusing to readers (I initially thought one of the 'L3' labels was a mistake and should have been 'L1' or 'L2'). Perhaps a short note in the caption could be added to clarify.

We updated the caption to clarify that L3 and L4 are branches at different depth levels.

Figure 3 caption - "Vertical dashed lines mark the temporal phases of seismicity (Fig. 2)." has been added to the 3c caption but I think it actually refers to panel d(?) Move this sentence to end of caption.

Corrected caption.

"Mikajima" -> "Miyakejima" (at the end of the second-to-last paragraph in section "Spatiotemporal evolution")

Corrected spelling to Miyake-jima (Japanese literature spelling).

Kind regards,

Diana Roman

Referee #2 (Paul Segall)

The revised manuscript addresses most of the issues raised in the initial reviews. In particular, the interpretation of the source region for the dikes is now made clearer. Also, the uncertainty in the depth of the deflating source, and its connection to the rest of the magmatic system has been clarified. While the paper is acceptable at this point, there are two points that I think could be improved.

Thank you again for your valuable review of our manuscript. We were thankful for your insightful and constructive input, which improved the scientific rigor of the manuscript.

In my initial review I noted “ Why should there be a threshold injection rate for eruption? The next sentence says the system hasn’t reached a threshold volume change. Physically, neither makes much sense – chamber overpressure relative to the local least compressive stress makes the most physical sense.” The authors refer to a paper by Browning et al (2015) to support the idea of an injection threshold and the magnitude of this threshold. I looked at this paper provides a “back of the envelope” calculation that makes rather strong, and possibly unrealistic, assumptions. I don’t think the authors are doing themselves a service to pin any interpretations on this estimate and would suggest omitting it entirely.

We agree that the study by Browning et al. (2015) makes a bold statement extending from chamber overpressure to chamber failure and eruption of Santorini. We removed the notion of a *threshold* and *minimal eruption-inducing volume*, as it is indeed too speculative and would require a more detailed discussion to be integrated in the context of the 2025 crisis. We only kept the reference and statement that the influx into the shallow Santorini chamber was apparently below dike- or eruption-inducing volumes.

The second point has to do with appealing to local and regional poroelastic stress changes to explain seismicity within the faulted basement. It is not clear what the authors mean here by “poroelastic”. Do you mean undrained poroelastic response, where there are stress changes induced by the dike opening, without any flow of pore-fluids in the basement rocks? Or are you envisioning that magmatic fluids are permeating out of the dike into the adjacent crust thereby altering the effective mean normal stress state? In the general theory of poroelasticity, changes in pore pressure induced deformations, and changes in deformation induce pore pressures. The usage in the manuscript is unclear.

During the previous revision, we thoroughly discussed this particular sentence. We see the shortcomings and have revised the sentence to be more specific: [...] *caused local and regional poroelastic stress changes due to opening and hydrothermal activation [...]*.

Response R2: Volcanic crisis reveals coupled magma system at Santorini and Kolumbo

As I said the paper is basically acceptable, so I leave it to the editors and authors to work out any further changes.

Paul Segall

This paper reports on a fascinating seismic sequence in the vicinity of Santorini, an active volcanic island in the Mediterranean, which will be of significant interest to Nature readers. The paper reports on new seismic, geodetic, and other observations which are relevant to understanding the process that gave rise to the sequence. In general, the paper is well written and suitable for publication. I do have a few suggestions which I believe will improve the manuscript.

The interpretation put forward by the authors is that there were a series of dikes propagating largely to the northeast starting somewhere below Santorini. This is supported by the earthquake locations and that limited GNSS data. The dike injections were preceded by inflation at Santorini and accompanied by subsidence measured by a sea-floor pressure gauge at Kolumbo volcano. This latter is interpreted as resulting from outflow from a shallow reservoir. It is not stated how well constrained the depth of this reservoir is, nor what features in the data control this part of the model. Is this feature controlled by the single pressure gauge? What controls its depth? The discussion section notes that there is deep seismicity below Kolumbo, however the connection between the subsidence and the deep seismicity is not clear and the depth separation is quite large.

Moreover, the origin of the magma for the dikes is unresolved. It does not appear to be from a shallow Santorini magma reservoir and the paper cites petrologic evidence for different source magmas at Santorini and Kolumbo. The schematic in Figure 4 shows an intermediate depth magma accumulation zone. The seismicity if anything seems to wrap around and above this zone. Is this the source of the magma? Elsewhere, in a quite speculative statement much of the seismicity is ascribed to dike induced pore-pressure changes in the surrounding crust. This and other statements suggest a significant role for tectonic faulting. While I understand that the authors do not have all the answers, I think it would be appropriate to more clearly indicate which aspects of the interpretation are strongest and where this considerable uncertainty, for example in the relative role of tectonic seismicity.

Smaller points that will clarify the manuscript.

Page 2, paragraph 2, refers to subsidence at Kolumbo crater but doesn't explain where that is located. Should refer to Figure 3 where it is identified.

Figure 3c, is supposed to show the temporal evolution of the dikes, here represented as rectangular dislocations. However, there is nothing to indicate the temporal progression. Could this be represented by color in some way, as is done in the supplement? The time

dependent dike model is, as I understand, a sequence of time intervals with best fitting rectangular dislocation. There is thus no physical connection between time steps. The smoothing is visually appealing but was not used in the modeling to predict deformation data.

Discussion First paragraph: Why should there be a threshold *injection rate* for eruption? The next sentence says the system hasn't reached a threshold *volume* change. Physically, neither makes much sense – chamber overpressure relative to the local least compressive stress makes the most physical sense.

Page 10 top The sentence “The dike-induced local stress changes induced fluid flow and pore pressure changes within the highly faulted basement of the Anhydros block and triggered secondary tectonic earthquakes, illuminating broad structural elements” is quite speculative. It may be plausible but, unless there is direct evidence for this it should be identified as speculation.

Figure 4a. needs reference for interpretation as mid crustal magma reservoir.

Paul Segall